# Silicon isotopic signatures of granitoids support increased weathering of subaerial land 3.7 billion years ago

Nicolas D. Greber [1,2,7] ✉, Madeleine E. Murphy[3,4,7], Julian-Christopher Storck [2], Jesse R. Reimink[5], Nicolas Dauphas[6] & Paul S. Savage[3]

The weathering and erosion of emerged land profoundly influences the Earth system, including the composition of the atmosphere and the type of nutrients delivered to the oceans. The emergence of land allowed for the formation of lakes and continental shelves, important habitats for the origin and evolution of life. Recent studies indicate a difference in silicon isotopes between Archean granitoids and their modern counterparts, which is explained by the incorporation of seawater-derived silica in the melting sources of the former. We show that this signature changed rapidly around 3.6 billion years ago, and that this shift is likely linked to an increase in the dissolved silicon flux from terrestrial weathering. Modeling suggests that the amount of oceanic silicon derived from terrigenous sources increased from near zero to around 32 ± 15% between 3.8 and 3.6 billion years ago. This indicates that, from this point onward, continental weathering feedbacks were established, and mass flux from land became an important source in the chemical budget of seawater, changes that likely exerted positive effects on the evolution of life.

Several lines of evidence suggest that between 25[1] and 75%[2] of continental crust already existed 3.0 billion years ago (Ga); however, it is unclear if this crust had emerged above sea level or was covered by epicontinental seas, and also how much earlier in Earth's history the continental crust started to become established. Continents impact the flow of nutrients to the ocean and allows for the existence of ponds, lakes and continental shelves[3]; such warm ponds and lakes are considered potential settings for the origin and evolution of life, as they likely contained high concentrations of essential biogenic elements[4–6]. Meanwhile, continental shelves provided shallow benthic zones and ecological niches that facilitated the diversification of microbial communities[7,8]. Furthermore, it is likely that even a small area of subaerial land would have a positive impact on a planet's habitability by strengthening climate weathering feedbacks[9,10]. In particular, weathering and erosion of emerged crust consumes $CO_2$, balancing its outgassing[11,12]. Below 1−1.5% subaerial land area, it is likely that silicate weathering of emerged crust becomes supply-limited, which describes a situation where weathering reactions run to completion due to a lack of weatherable rocks[9–11]. This condition prevents an efficient and continuous weathering feedback between atmosphere and land, making

seafloor weathering the dominant factor to balance the carbon cycle[11]. Once the area of land overcomes this threshold, continental weathering becomes the primary driver of $CO_2$ regulation at steady-state[11]. This bears importance, as a better understanding of the weathering feedback mechanisms of the early Earth enhances our understanding of the environmental conditions that supported the development of early life[11].

While there is no agreement on the precise timing, geochemical evidence and zircon age distributions point to a rise in the surface area of emerged crust on Earth between 3.0 Ga and 2.5 Ga[13–17], with land exposure eventually approaching its present value between 2.8 Ga and 2.5 Ga[16,17]. However, the observation of ca. 3.7 to 3.8 Ga clastic sediments[18,19], light oxygen isotopes in zircon[20,21], the Sr isotopic composition of 3.52 to 3.20 Ga baryte[22], Ge/Si ratios in banded iron formations[23], and a machine learning approach based on a global dataset of basaltic rocks[24] support the view that significant emerged land has existed since 4.0 Ga to 3.7 Ga. If true, the ability of the Earth system to mediate its atmospheric $CO_2$ by chemical weathering of emerged land, as happens on modern Earth, may have occurred relatively early in our planet's history.

[1]Muséum d'histoire naturelle de Genève, Genève, Switzerland. [2]Institute of Geological Sciences, University of Bern, Bern, Switzerland. [3]School of Earth and Environmental Sciences, University of St Andrews, St Andrews, Scotland, UK. [4]Lamont-Doherty Earth Observatory, Columbia University, New York, NY, USA. [5]Department of Geosciences, Pennsylvania State University, university park, PA, USA. [6]Origins Laboratory, Department of the Geophysical Sciences and Enrico Fermi Institute, The University of Chicago, Chicago, IL, USA. [7]These authors contributed equally: Nicolas D. Greber, Madeleine E. Murphy. ✉e-mail: nicolas.greber@geneve.ch

Current research shows that the heavy Si isotopic composition ($\delta^{30}Si$) of granitoid rocks and mineral separates from the early Archean (since ~3.8–4.0 Ga) reflects the presence of silicified oceanic crust in their melt sources[25–28]. The isotopic composition of silica deposited on the seafloor during the Precambrian (e.g., in chert) is influenced by the $\delta^{30}Si$ of seawater, which is largely, though not entirely, controlled by the flux of Si released into the dissolved pool during the chemical weathering of continental landmasses[29,30]. This is because clay formation from chemical weathering of continental crust *sensu lato* enriches the dissolved pool in heavy Si isotopes while creating an isotopically light reservoir in the form of clay-rich soils and sediments[31,32]. Current estimates suggest that today around 60 to 70% of the flux of Si to the modern oceans is directly derived from chemical weathering of continental crust via rivers, with a further ~25% from the dissolution of continental dust and sediments[31,33]. The $\delta^{30}Si$ of Archean granitoids may therefore indirectly reflect the existence and degree of subaerial weathering of land on Earth at or before the age of granitoid formation.

The oldest preserved granitoids that formed by fractional crystallization of hydrous melts are about 3.9 billion years old[34,35]. These melts either formed by partial melting of subducted oceanic crust, or by partial melting of hydrated metabasalts brought to depth by vertical tectonics[36]. Therefore, the age of the granitoids postdates the silicification event of the seafloor. In the case of subduction, the speed of tectonic plates scales with the rate of heat loss, which was between similar[37] to three times higher[38] than present. This makes the age of the oldest modern oceanic crust of around 200 Ma[39] a maximum estimate for the time difference between the silica addition to the seafloor and the formation of granitoids. As vertical tectonics is thought to require less time than that to bury material past its solidus[40–42], the time it took from the silicification of the seafloor to its melting and granitoid formation was on average likely less than 200 Ma, independent of the tectonic regime. Studying the Si isotopic composition of ancient granitoids could thus provide insights into the global surface cycle of Si at the Hadean-Archean boundary, or even before.

We therefore measured the Si (and Ti) isotopic compositions of 4.02 to 2.94 Ga rocks from Greenland and the Canadian Acasta Gneiss complex and show that the Si isotopic composition of granitoids increased geologically rapidly around 3.6 Ga ago. We interpret this abrupt shift in the isotopic composition as reflecting an increase in the Si flux from the emerged crust to the oceans at 3.8 to 3.6 Ga.

## Results

Sample descriptions are given in supplementary Note 1, and the results are shown in supplementary Data 1. The 25 analyzed samples include: (i) seven 4.02 Ga to 3.93 Ga tonalites that formed via fractional crystallization of dry melts extracted from a primitive mantle source[43,44], and (ii) 18 granitoids ranging in age from 3.80 Ga to 2.94 Ga that are the product of fractional crystallization of hydrous melts originating from the partial melting of a hydrated oceanic crust. For 17 of the samples, Ti isotopic compositions are already published[44]; for the rest we performed new Ti analyses. Titanium isotopic data are reported as $\delta^{49}Ti$, the permil deviation of the $^{49}Ti/^{47}Ti$ ratio relative to OL-Ti standard. The stable Si isotope data is presented as $\delta^{30}Si$ (permil deviation in the $^{30}Si/^{28}Si$ ratio relative to NBS28) and $\Delta^{30}Si$. This latter value is the difference between the measured $\delta^{30}Si$ and that predicted based on the modern igneous array: $\Delta^{30}Si = \delta^{30}Si_{measured} - ((SiO_2*0.0056) - 0.567)$ [45]; this corrects for the effect of fractional crystallization on Si isotopes, which is primarily a function of a melt's $SiO_2$ content. A significant deviation from zero in $\Delta^{30}Si$ thus implies non-igneous sources in the melt (Fig. 1). Note, this should not be mistaken for the notation $\Delta^{'29}Si$ used to quantify the

**Fig. 1 | Silicon and Ti isotope data from this study.**
**A** Silicon isotopic composition of measured granitoids compared to their $SiO_2$ concentrations. The gray bar illustrates the correlation between these parameters of modern igneous rocks[45]. The oldest samples follow the trend defined by modern rocks, whereby samples younger than 3.8 Ga are on average enriched in $^{30}Si$. **B** Titanium isotopic composition versus $SiO_2$ concentration of Archean rocks from this study and from the literature[34,35,44,96]. Indicated with arrows are the approximate trends of modern intraplate magmatic systems (e.g., hotspot volcanism akin to Hawaii) and subduction related magmatism (e.g., Aegaean Arc). One sample (155768) deviates from the trend of the other rocks, but does not exhibit an aberrant Si isotopic composition. **C** Silicon versus Ti isotopic compositions of granitoids from this study. Granitoids ≤3.8 Ga have lighter Ti isotopic compositions, but variably enriched Si isotopic signatures compared to the older samples. **D** Comparison of the Ti isotopic magmatic index (TIMI[44]) with $\Delta^{30}Si$ value, representing the offset between the measured data and that expected from the modern igneous array (see **A**). Fractionation products of dry melts as indicated by TIMI ≥ 1 have $\Delta^{30}Si$ around 0, fractionation products of hydrous melts (TIMI ≤ 0.85) originating from the partial melting of hydrous basalts have $\Delta^{30}Si > 0$. Rocks older than 3.6 Ga (orange circles) have lower $\Delta^{30}Si$ than younger samples (blue squares). Open symbols are average values ± 2SE. Errors on individual data are 2 SD. Data for these Figures are shown in supplementary Data 1 and 2.

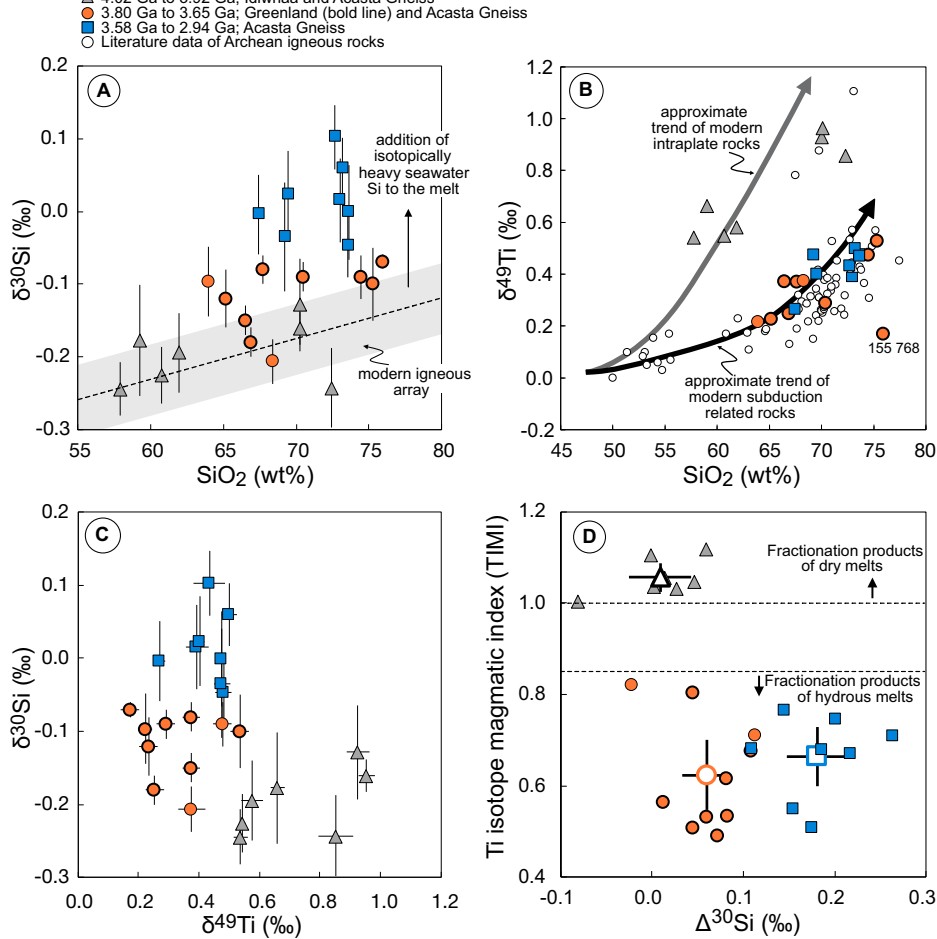

departure from a reference mass-fraction law when 3 Si isotopes are measured[46], or an isotopic fractionation factor.

The 3.93 to 4.02 Ga samples that derive from the fractional crystallization of dry magmas exhibit a $\Delta^{30}Si$ of zero ($+0.009 \pm 0.034$; 2SE, $n = 7$) (Fig. 1). Their Si isotopic compositions thus follow the same fractionation trend as observed in Phanerozoic systems (Fig. 1A), and resemble modern I- and/or A-type granites, in agreement with an earlier study about similar rocks from that region[28]. In contrast, the granitoids with ages ≤3.80 Ga have

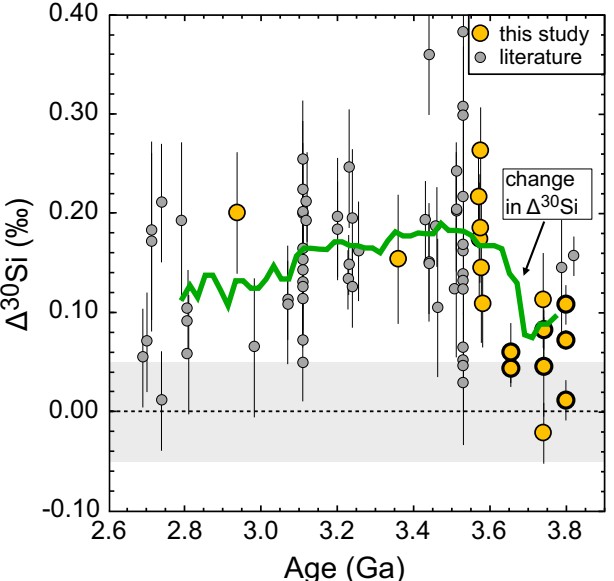

**Fig. 2 | Silicon isotopic offset ($\Delta^{30}Si$) of Archean granitoids from modern equivalents through time.** Large, yellow circles are samples from this study, small, gray circles are from literature[25–27]. The green line is the mean value calculated each 2 Ma for a ± 100 Ma age range. Visible is a rapid change in the $\Delta^{30}Si$ value of the granitoids at around 3.6 Ga, followed by a gradual decrease towards younger samples. Samples from this study with a bold symbol outline are from Greenland, the others originate from the Acasta Gneiss. The gray bar around 0‰ is the ±0.05‰ uncertainty on the modern igneous array[45]. Errors for data from this study are 2 SD, errors for literature data are as published in the original source. For references see supplementary Data 1 and 3.

overall positive $\Delta^{30}Si$, indicating that isotopically heavy Si was added to their melting source[25–27] (Fig. 1).

The Ti isotopic composition of igneous rocks can be used to trace if an intermediate to felsic magmatic rock originated from the fractional crystallization of a dry or hydrous melt[44]. The $\delta^{49}Ti$ of the seven 3.93 Ga to 4.02 Ga samples from group (i) have previously been attributed to the fractionation of dry magma, as opposed to the melting of hydrated oceanic crust[44], in agreement with other evidence[47]. In contrast, all samples from group (ii) have $\delta^{49}Ti$ indicative of fractional crystallization of a hydrous melt (Fig. 1 and supplementary Data 2), originating from the melting of metabasalt. The difference in the $\Delta^{30}Si$ between group (i) and (ii) is thus likely due to the distinct genesis of these rocks.

Within the granitoids that derive from the melting of a hydrous source, the $\Delta^{30}Si$ values exhibit an increase at 3.60 Ga (Fig. 1) from $+0.060 \pm 0.026‰$ to $+0.181 \pm 0.033‰$ (both 2SE; $n = 10$ and $n = 8$), a pattern that remains when bulk rock literature data are included[25–28] (Fig. 2 and supplementary Data 3). Note that the $\delta^{30}Si$ signatures of the granitoids also exhibit a shift towards more positive values at 3.60 Ga (supplementary Fig. 1). Taking all available data into account, granitoids with ages between 3.82 Ga and 3.65 Ga have a $\Delta^{30}Si$ of $+0.075 \pm 0.030‰$ (2SE, $n = 12$), which is significantly lower compared to samples that are younger than 3.60 Ga, yielding $+0.160 \pm 0.018‰$ (2SE, $n = 67$). Furthermore, after the $\Delta^{30}Si$ value peaks at around 3.50 Ga, the Si isotopic composition seems to gradually decrease to around $+0.12‰$ towards the end of the Archean (Fig. 2), making the shift in the $\Delta^{30}Si$ at around 3.60 Ga even more notable.

## Discussion

### Rapid change in Archean granitoid Si isotopic composition due to an increase in, or the onset of silica input to the oceans from emerged crust

Models of Si isotopic fractionation during melting and crystallization are unable to completely account for the heavy Si isotopic compositions of Archean granitoids[25–27], indicating that it is likely a source signature, reflecting the silicification of the oceanic crust through seawater circulation. Terrestrial oceans are thought to have existed since at least 4.3 Ga[48,49]. In the absence of silicifying organisms, seawater was likely nearly saturated with Si throughout the periods corresponding to the rocks studied here[50]. Thus, the observed shift in $\Delta^{30}Si$ in Archean granitoids at 3.6 Ga most likely reflects a change in the silicon cycle on a global scale, through either an increase in the proportion of marine silica incorporated in granitoid melts, or a change in

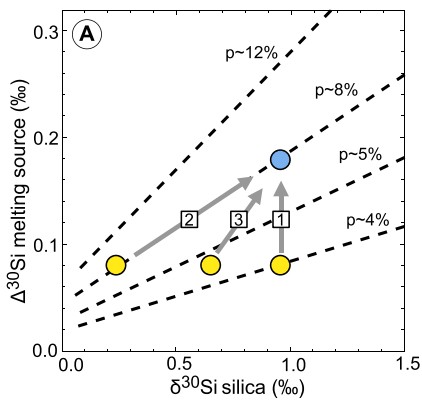

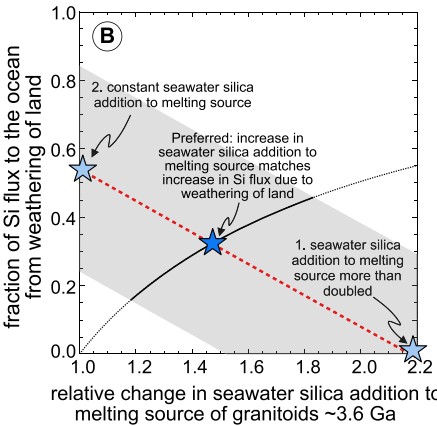

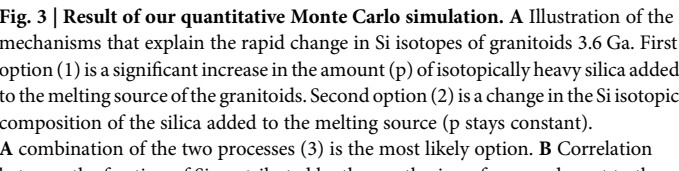

**Fig. 3 | Result of our quantitative Monte Carlo simulation. A** Illustration of the mechanisms that explain the rapid change in Si isotopes of granitoids 3.6 Ga. First option (1) is a significant increase in the amount (p) of isotopically heavy silica added to the melting source of the granitoids. Second option (2) is a change in the Si isotopic composition of the silica added to the melting source (p stays constant). **A** combination of the two processes (3) is the most likely option. **B** Correlation between the fraction of Si contributed by the weathering of emerged crust to the

ocean and the increase in seawater silica incorporated into the melting source of granitoids that explains the shift in their Si isotopic composition at 3.6 Ga (see also supplementary Fig. 3 for the raw result of the Monte Carlo model). The dotted red line represents the average, the gray field indicates the 95% confidence level, and the black line shows the solution for our preferred scenario, where the increase in the Si flux to the ocean from the weathering of land is linearly correlated with the increase in seawater Si in the melting source of the granitoids.

the $\delta^{30}$Si of silica added to the altered oceanic crust (Fig. 3), or a combination of both. We argue that the best explanation for the change in $\Delta^{30}$Si of granitoids at 3.6 Ga is a surge in, or even the onset of, subaerial weathering of land and concomitant (isotopically heavy) silicon discharge to the oceans.

An increase in the average degree of silicification of the seafloor can only be achieved by a substantial augmentation of the silica flux into seawater. Without land, the only major silica source for the oceans is hydrothermalism[29,30,51]. However, there is no reason to infer a sudden increase in hydrothermal input at this time; infact, outgassing and hydrothermal circulation likely has become less intense over Earth history[52–54], meaning that its contribution to the silica flux likely decreased rather than increased. A change in the proportion of the oceanic silicon sinks like reverse weathering (consumption of dissolved seawater cations, as e.g., $Si^{4+}$, to form clay minerals and release $CO_2$[55,56]), or in the magnitude of Si isotope fractionation during precipitation of chert due to a cooling ocean could influence seawater $\delta^{30}$Si as well. However, no sudden change in these parameters are documented between 3.8 and 3.6 Ga, and it is generally thought that they have changed in a gradual manner[51,56–58] and therefore cannot explain a rapid shift in the Si isotopic composition of seawater. Additional cations supplied by river runoff from the chemical weathering of land may have enhanced the formation of authigenic marine clays and thus the release of heavy Si isotopes into seawater[59,60]. However, their impact on the Si isotopic composition of seawater is uncertain, especially since reverse weathering in the early Archean ocean was likely widespread[56], utilizing cations that also originated from non-continental sources, such as $Fe^{2+}$ [56]. Therefore, dissolved, isotopically heavy Si in river runoff was likely the dominant factor that led to an increase in the $\delta^{30}$Si of seawater around 3.8 to 3.6 Ga.

Continental river runoff from emerged lands increases the $\delta^{30}$Si of seawater, because Si isotope fractionation associated with the formation of clay minerals by chemical weathering leads to an isotopically heavy, dissolved Si pool transported to the oceans by rivers[31,32], and an isotopically light pool of clay minerals and iron oxides[61], which is (at least initially) sequestered in emerged lands and the continental shelf. Dissolved riverine Si is the largest modern input to the oceans and has a $\delta^{30}$Si of $+1.21 \pm 0.76$‰, significantly heavier than that of the hydrothermal flux ($\delta^{30}$Si $= -0.30 \pm 0.14$‰)[62]. Therefore, the onset of alteration of crust exposed to the atmosphere would add, on a relatively rapid timescale, a significant amount of isotopically heavy Si to the ocean, assuming chemical weathering and desilicification of crustal igneous material occurred via similar mechanisms as today.

This assumption bears discussion. There is a smaller Si isotope fractionation associated with the formation of poorly-desilicified clays compared to those which have undergone more intense chemical weathering (smectite vs. kaolinite)[63]. Thus, a difference in clay mineral formation in the Archean compared to today could have impacted riverine Si isotopic compositions. The type of clay minerals formed during the weathering of land in the early Archean remains poorly constrained, partly due to limited knowledge of the conditions of alteration. However, models of chemical weathering in the late Archean under anoxic conditions and higher $CO_2$ concentrations suggest that a mix of kaolinite and smectite would be common weathering products and that the average silica concentration in river water remained similar over time[64]. Kaolinite is a common product of chemical weathering of feldspar and biotite under modern acidic conditions[65,66] and shows strong negative Si isotope enrichment[32]. Furthermore, Archean shales which partly include material altered on land are also reported to have comparably light Si isotopic compositions to their modern counterparts[67,68], indicating that chemical weathering at that time also produced clay minerals incorporating isotopically light Si.

Many factors influence Si isotope fractionation during chemical weathering of land and thus the $\delta^{30}$Si value of river water. These factors include the type of clay minerals that are initially formed[63], their potential transformation during prolonged alteration, whether weathering reactions occur under kinetic or equilibrium conditions[69], and whether or not Si gets adsorbed on Fe-hydroxides[68,70]. These chemical weathering reactions occur on modern emerged crust and under a range of pHs, resulting in river water

transporting a heavy Si isotopic signature. It is thus difficult to precisely predict the Si isotopic composition of an early Archean river runoff, but altogether and given our current understanding of early Archean processes, it is likely that rivers at that time also carried isotopically heavy silicon.

A sudden, strong increase in the flux of dissolved silicon may also explain the apparent slow decrease in granitoid $\Delta^{30}$Si after having reached a peak at around 3.5 Ga (Fig. 2). As weathering of land transports heavy Si isotopes in river water to the ocean, a complementary, isotopically light reservoir forms in subaerial basins and the continental shelf. While a portion of this isotopically light material is deposited as shales in the ocean[67], some of its silica content is anticipated to undergo mobilization either through the dissolution of aeolian, riverborne or continental shelf sediments[31], or via the leaching of iron-bearing minerals[68]. The transitional period during which this reservoir of isotopically light material on the emerged crust gradually increased would be accompanied by a temporary augmentation in the $\delta^{30}$Si flux to the oceans, persisting until a new equilibrium was reached. This new equilibrium corresponds to a Si isotope flux from land to the oceans which is buffered by continentally-derived pelitic sediments. The leaching of isotopically light Si from the pelites mixes with the isotopically heavy riverine runoff, resulting in a $\delta^{30}$Si of the oceans (and the silicified oceanic crust) that is heavier than before the onset of continental weathering, but lower than during the transitional period.

High-temperature igneous and metamorphic processes can generate isotope fractionation in both the Si and Ti isotope systems too. However, it is unlikely that a changing melting regime (pressure and degree of partial melting) significantly altered the proportion or isotopic composition of marine silica incorporated into granitoid melts. This is because we do not observe any systematic differences in the ≤3.80 Ga samples in chemical indices that are classically argued to be sensitive to melting pressure (e.g., La/Yb, Sr/Y, Ta vs. Nb concentrations)[36], or degree of partial melting ($K_2O$/$Na_2O$)[71] (see supplementary Fig. 2 and supplementary Data 1).

It has also been shown that granulite xenoliths have the same Si isotopic composition as their inferred unmetamorphosed protoliths, indicating a limited effect of metamorphism on Si isotopes of igneous rocks[72]. Furthermore, the fact that samples from the Acasta Gneiss and from Greenland with ages between 3.8 and 3.6 Ga exhibit, on average, a positive but low $\Delta^{30}$Si signature suggests that metamorphic overprinting is unlikely to be the cause for the change in the Si isotopic composition of granitoids at 3.6 Ga. Firstly, this would require that the different peak metamorphic conditions experienced by the samples from Acasta and Greenland had an identical effect on their Si isotopic compositions, which seems unlikely. Secondly, the Acasta Gneiss samples ranging in age from 3.74 to 2.94 Ga exhibit the same trend in $\Delta^{30}$Si as the combined sample set from Greenland (this study) and literature data that mainly derives from South African rocks[27] (Fig. 2). To produce a change at 3.6 Ga in the $\Delta^{30}$Si within the Acasta Gneiss, metamorphism after 3.75 Ga would have had to affect the Si isotopes differently in rocks of different ages but from the same complex. Thus, in contrast to an increase in the silica runoff or the onset of weathering of emerged crust, a changing melting regime or a metamorphic effect are unlikely to be the cause for the shift in $\Delta^{30}$Si of granitoids at 3.6 Ga.

Based on trace elements (La/Yb) and radiogenic isotope signatures ($\epsilon$Hf and $\mu^{142}$Nd), it is thought that the Acasta gneisses with ages between 3.96 and 3.75 Ga formed via shallow melting of mafic crust of Hadean age (4.20 to 4.30 Ga)[43]. In contrast, rocks with ages younger than 3.60 Ga likely originate from the melting of younger crust at higher pressures[43], a change that temporally coincides with the observed increase in $\Delta^{30}$Si. It has been suggested that this change in the radiogenic isotopes in the Acasta Gneiss complex (and other locations) between 3.75 and 3.60 Ga is due to the onset of horizontal or mobile-lid tectonics[43,73], pointing to an eventual link with the increase in $\Delta^{30}$Si of granitoids. However, whether the emergence of land and the tectonic regime are interlinked remains a topic of debate[1,10,74,75], with some suggesting that the cycling of clastic sediments may have stabilized subduction processes[76] or facilitated continental emergence independent of the geodynamics[77]. Our findings on the increased input of dissolved Si indicate that chemical weathering of newly emerged land played a

significant role around 3.8 to 3.6 Ga. However, it remains unclear whether this was accompanied by substantial physical erosion and a significant discharge of clastic sediments into the oceans, leaving open the question as to whether shifts in terrestrial runoff are linked to changes in the tectonic regime.

## A continental silicon runoff modeling approach

We explore here quantitative mass balance models that can explain the shift in $\Delta^{30}Si$ of Archean granitoids from +0.075‰ to +0.160‰ at around 3.6 Ga. These models consider either a change in the quantity of silica added to the melt source, a change in the $\delta^{30}Si$ of seawater-derived silica added to the melt source, or a combination of both (Fig. 3; details of the models are given in supplementary Note 2, and input parameters are summarized in supplementary Data 4).

The two first scenarios (a change in the quantity of silica added to the melt source, or a change in the $\delta^{30}Si$ of seawater-derived silica added to the melt source) are endmembers (Fig. 3). More realistically, weathering of land increased both the Si isotopic composition of seawater and the Si flux to the ocean. It is difficult to assess exactly how the addition of Si from the weathering of emerged crust to an already nearly silica saturated ocean[50] impacts the amount of seawater silica incorporated into the melting source of the granitoids. We thus have developed a model that combines the effect of an elevated $\delta^{30}Si$ signature due to the weathering of emerged land and a higher contribution of seawater silica to the melting source of the granitoids (Model 3 in the supplementary Note 2). This model is based on two assumptions; (i) the change in the isotopic composition of the silica added to the melting source of the granitoids is equal to the change in seawater $\delta^{30}Si$ due to the onset of the weathering of land, and (ii) the $\Delta^{30}Si$ of the granitoids with ages between 3.82 Ga and 3.65 Ga (i.e., +0.075 ± 0.030‰) does not require a silica flux to the ocean from emerged crust. The main Si sinks in the early Archean ocean were likely amorphous silica precipitation and reverse weathering, which exhibit isotope fractionations with seawater of around −0.5‰ and −2.0‰, respectively[78–80]. A $\Delta^{30}Si$ of +0.075 can be explained with a seawater $\delta^{30}Si$ if reverse weathering makes up 10% to 20% of the Si sink, which is in agreement with estimates for the early Archean, albeit with large errors[51,56,81] (see supplementary Note 2).

The results of the models show that if the seawater silica added to the melting source approximately doubled, then no change in the seawater $\delta^{30}Si$ is required (case 1 in Fig. 3). To explain this scenario requires either that this additional Si did not derive from the weathering of emerged crust, or that the Si isotopic composition delivered by land to the ocean had a much lower Si isotopic composition than in the modern, more akin to that of the unweathered continental crust. The latter is more likely, as increasing seawater silica in the melting source of granitoids requires a higher flux of Si to the ocean. There is currently no documented evidence of a sudden increase in hydrothermal discharge between 3.8 Ga and 3.6 Ga. Instead, it is generally suggested that volcanic outgassing decreased gradually[52–54], and aside from Si input from emerged land, no other major Si flux to the ocean is known.

In contrast, if the amount of seawater silica added to the melting source of the granitoids remained constant over time, then the $\Delta^{30}Si$ of post-3.6 Ga granitoids would require an isotopically heavy Si flux from emerged land making up 53% ± 26% of the total Si input to the ocean (case 2 in Fig. 3). However, this is likely an overestimate as the silicon contributed to seawater through chemical weathering of land must be balanced by a greater Si output.

Therefore, our preferred scenario is when the proportion of the Si input from the emerged crust (with heavy $\delta^{30}Si$) is equivalent to the increase in the amount of Si added to the melting source of the granitoids, meaning that these two parameters are linearly correlated (case 3 in Fig. 3). In this case, the model predicts that 32 ± 15% of the Si that was added to the seafloor to produce the post 3.6 Ga granitoids originated from the weathering of land (with the rest derived from hydrothermalism).

Estimating early Archean hydrothermal activity is challenging, as it is connected to the relationship between mantle temperature and convective heat transport which is poorly understood[82]. Generally, the hydrothermal

heat flux should scale as the square of the total heat flux. However, estimates of the total heat flux over time vary significantly, ranging from little change since 4.0 Ga[37] to being up to three times higher[38]. Furthermore, the hydrothermal flux of Si to the Archean oceans was likely higher than it is today as its Si concentration was possibly up to four times more concentrated than the current levels[83]. Despite these uncertainties, it has been estimated that the hydrothermal Si flux to the oceans at 3.5 Ga was, at most, five times higher than today[56] and that the hydrothermal Fe flux at 3.8 Ga was up to 8.5 times higher than modern levels[54]. As this latter study did not take into account the potentially lower Fe concentrations in early Archean hydrothermal fluids[83] it can be used as an analogue for the Si flux. Therefore, with both higher concentrations and higher flux, conservative estimates of the hydrothermal Si input at 3.8 Ga range from 5 to 34 times that of the present. If we take the results of our model as is, i.e., that the hydrothermal Si input accounted for 68% of the total Si flux to the ocean, then our calculations suggest that the Si supply from land after 3.6 Ga could have been 0.2 to 4.4 times that of the present in absolute numbers, depending on the chosen values for the hydrothermal Si input in the modern (i.e., between 0.6 and 1.7 Tm/yr[31,33]) and assuming a modern dissolved riverine Si input of 6.2 Tm/yr[31,33]. Given all the uncertainties, this describes a plausible range for a terrestrial silica runoff 3.6 to 3.8 Ga.

## Establishing weathering feedbacks between atmosphere and land on Earth

As argued earlier, the silicified material that gave rise to the $\Delta^{30}Si$ of the post-3.6 Ga granitoids could have been added to the oceanic crust up to 200 Ma before granitoid formation. Our data therefore suggest that prior to the generally accepted large-scale increase in the areal extent of land between 3.0 Ga and 2.5 Ga[13–17], weathering of emerged crust was already occurring between 3.8 Ga and 3.6 Ga. This aligns temporally with other evidence suggesting that some emerged land existed in the early Archean[21–24], and the appearance of detrital sediments and banded iron formations around 3.7 to 3.8 Ga[18,19,84]. Both of these lithologies necessitate land and continental margins for their formation. The emergence of land may thus have occurred in a stepwise manner, with an initial phase around 3.8 to 3.6 Ga and a second, more pronounced increase in subaerial land around 3.0 to 2.5 Ga.

The weathering and erosion of the emerged crust consumes $CO_2$, which helps to balance $CO_2$ outgassing[11,12]. When total subaerial land area is sufficiently small, continental weathering becomes supply limited, meaning that weathering reactions run to completion and do not enable the development of a continuous weathering feedback between atmosphere and land[10]. Once the areal extent of land is large enough to surpass the supply limited weathering regime, a steady-state establishes between the degassing of $CO_2$ and the scale of continental weathering[11]. From this point on, it is likely that the extent of continental weathering is predominantly controlled by $CO_2$ outgassing and not by the surface area of land. Our model implies that the continental silicon flux to the oceans before around 3.8 Ga to 3.6 Ga was low, but then increased rapidly so that the flux from emerged land contributed 32 ± 15% to the total input by the end of this period. It is thus likely that since this time, an efficient weathering feedback mechanism between atmosphere and land existed on Earth. This timeframe is later, however, than the finding of light O isotopes in 4.0 Ga zircon that seemingly requires interaction between fresh water and emerged land[20]. However, these data are isolated and appear punctually, making their relevance for global weathering feedbacks ambiguous, as they could also be explained with transient subaerial crust. Furthermore, an O isotopic composition of zircon of approximately ≥2‰ as observed for the samples older than 3.7 Ga[20] might also be due to melting of hydrothermally altered crust[85].

The finding that after 3.8 to 3.6 Ga around a third of the total Si flux to the ocean was provided by the continental runoff shows that, at least from this point in Earth history onwards, emerged land had a major impact on the cycling of elements, the flow of nutrients to the oceans, and the stabilization of the climate through weathering feedbacks. These parameters are thought to positively impact the development of life, with microbial communities able to form stromatolites as early as 3.5 Ga[86,87].

## Methods

### Silicon isotope measurements

The Si isotopic compositions were determined in the St Andrews Isotope Geochemistry (STAiG) laboratories at the University of St Andrews. Analytical procedures followed those described in Murphy et al.[25], which are based on the methods of Georg et al.[88]. In brief, ~10 mg of each sample powder was dissolved via alkali-fusion, with ~200 mg superconductor-grade NaOH as the flux. The flux and sample powder were fused at 720 °C for around 15 min. Then, the fusion cake was quenched in MQ-e water and transferred from the crucible to pre-cleaned PP sample bottles via a pipette. Each sample was diluted further with MQ-e water to reach a Si concentration of ~10−20 ppm Si, and was then acidified to 0.23 M $HNO_3$. The final Si concentration of each sample was analysed using photospectrometry, to check for fusion yield and for column chemistry loading volumes. Silicon was purified from the sample matrix using a single-stage column procedure, using an AG50W-X12 BioRad cation exchange column.

Silicon isotopes were measured on the Neptune Plus MC-ICP-MS in the STAiG labs in St Andrews. All three Si isotopes were measured simultaneously, and the instrument was operated in medium resolution mode ($M/\Delta M \sim 7000$) to allow for discrimination between the silicon isotope beams and significant molecular interferences. Sample isotope compositions were calculated via standard-sample bracketing, using the NBS28 standard, and with on-peak blanks bracketing every measurement. Blank corrections and delta values ($\delta^{30}Si$ and $\delta^{29}Si$) were calculated off-line, and the delta values were checked for mass-dependence for data quality. Each sample was analysed at least 3 separate times over the course of an analytical session, and the error is calculated as 2 standard deviations of the mean of these data. For further information, please see Murphy et al.[25].

The standards BHVO-2, BIR-1, GSP-2 and Diatomite were analysed at the same time as the samples to check for method accuracy. These data can be found in supplementary Data 1 and are identical within error to reference values[25,89,90].

### Titanium isotope measurements

The Ti isotopic compositions were determined at the University of Bern, following established protocols[62,91]. In brief, powdered samples were fused with $LiBO_3$ at 1100 °C for approximately ten minutes. The resulting glass pellets were fragmented, and clean aliquots containing approximately 3 μg of Ti were weighed into Savillex beakers. These aliquots were then mixed with a $^{47}Ti$–$^{49}Ti$ double spike. Subsequently, the spiked samples underwent complete digestion in 3 M $HNO_3$ and were then dried down to reach sample-spike equilibration. The samples were then dissolved in 12 M $HNO_3$, and Ti was isolated from the sample matrix through a two-step ion-exchange chromatography process. This process involved initial separation of Ti on a DGA column (TrisKem International), followed by purification on an AG1-X8 Bio-Rad column.

Titanium isotopic compositions were analyzed using a Neptune plus MC-ICP-MS, simultaneously measuring Ti isotopes at masses 46, 47, 48, 49, and 50, along with $^{44}Ca$. Data reduction was conducted offline, encompassing on-peak zero correction, interference correction, double spike deconvolution, and standard sample bracketing, following published procedures[92].

Geostandards G-2 and RGM-1 were processed and analyzed alongside the samples. The $\delta^{49}Ti$ values obtained were +0.46 ± 0.03‰ and +0.55 ± 0.03‰, respectively, consistent with published data[93,94] (supplementary Data 1). The 2 SD uncertainty of a measurement is estimated based on the long-term 2 SD reproducibility of various geostandards at the University of Bern, yielding an uncertainty of ±0.03‰[95].

## Data availability

All data is also available at Mendeley Data at https://data.mendeley.com/datasets/fsstf6kj76/1.

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

## Acknowledgements

This project was founded by the Swiss National Science Foundation grant 181172 to N.D.G. The Neptune MC-ICP-MS at the University of Bern was acquired with funds from the National Centre for Competence in Research PlanetS, also supported by the Swiss National Science Foundation (SNSF). M.E.M. was supported by the University of St Andrews School of Earth and Environmental Sciences and the Handsel scheme, in addition to NERC grant NE/R002134/1 to P.S.S. We thank three anonymous reviewers for their constructive criticism and the associate editor for the handling of the manuscript. We are grateful to Stephen Moorbath, Robert Pankhurst, the Geological Survey of Greenland, and all others involved in collecting the Greenland samples. We also thank the Northwest Territories Geoscience Office staff for their invaluable logistical and scientific support during several field campaigns working in the Acasta Gneiss Complex.

## Author contributions

N.D.G. contributed to the conceptualization of the project, the formal analysis, the visualization, the writing of the original manuscript and helped with funding acquisition. M.E.M. contributed to the conceptualization, Si and Ti isotope measurements, and writing of the original manuscript. J.C.S. contributed to the Ti isotope measurements and the reviewing and editing of the manuscript. J.R.R. provided sample material and helped with the reviewing and editing of the manuscript. N.D. contributed to the conceptualization of the project and the reviewing and editing of the manuscript. P.S.S. contributed to the conceptualization of the project, provided samples from Greenland, helped with funding acquisition and the Si measurements, and the writing of the original manuscript.

## Competing interests

The authors declare no competing interests.
