## [Transparent Peer Review file · Communications Earth & Environment]

Silicon isotopic signatures of granitoids support increased weathering of subaerial land 3.7 billion years ago

Corresponding Author: Professor Nicolas Greber

Version 0:

Decision Letter:

Dear Professor Greber,

Your manuscript titled "Silicon Isotopic Evidence for an Intensification in Continental Weathering 3.7 Billion Years Ago" has now been seen by 2 reviewers, whose comments are appended below. You will see that they find your work of some potential interest. However, they have raised quite substantial concerns that must be addressed. In light of these comments, we cannot accept the manuscript for publication, at least in its present form.

We hope you will find the reviewers' comments useful as you decide how to proceed. Should you decide to address these criticisms, please ensure that the following editorial thresholds are met:

**** Provide justification for the calculated silica flux provided by early land surfaces as mentioned by the first reviewer.**

**** Provide an explanation for the suggested silica flux from hydrothermals as pointed out by the first reviewer.**

**** Clarify the uncertainty associated with the estimated silica fluxes as suggested by the second reviewer.**

If you choose to re-submit your manuscript, please either highlight all changes in the manuscript text file or provide a list of the changes made to the manuscript along with your point-by-point response to the reviewers' concerns.

If the revision process takes significantly longer than three months, we will be happy to reconsider your paper at a later date, as long as nothing similar has been accepted for publication at Communications Earth & Environment or published elsewhere in the meantime.

Please use the following link to submit your revised manuscript, a point-by-point response to the reviewers' comments with a list of your changes to the manuscript text (which should be in a separate document to any cover letter), a tracked-changes version of the manuscript (as a PDF file) and any completed checklist:

Link Redacted

Please do not hesitate to contact us if you have any questions or would like to discuss the required revisions further. Thank you for the opportunity to review your work.

Best regards,

Mojtaba Fakhraee, PhD
Editorial Board Member
Communications Earth & Environment
orcid.org/0000-0002-2461-6374

Carolina Ortiz Guerrero, PhD
Associate Editor
Communications Earth & Environment

EDITORIAL POLICIES AND FORMAT

If you decide to resubmit your paper, please ensure that your manuscript complies with our editorial policies and complete and upload the checklist below as a Related Manuscript file type with the revised article:

Editorial Policy Policy requirements
(Download the link to your computer as a PDF.)

- Behavioural and social science
- Ecological, evolutionary & environmental sciences
- Life sciences

<https://www.nature.com/documents/nr-reporting-summary.zip>

For your information, you can find some guidance regarding format requirements summarized on the following checklist: (<https://www.nature.com/documents/commsj-phys-style-formatting-checklist-article.pdf>) and formatting guide (<https://www.nature.com/documents/commsj-phys-style-formatting-guide-accept.pdf>).

REVIEWER COMMENTS:

Reviewer #1 (Remarks to the Author):

The authors demonstrate a change in the Si isotopic composition ($\delta^{30}\text{Si}$) of granitoids between 4.0 and 3.5 Ga from about -0.25 per mil to +0.1 per mil. This evolution is explained as reflecting a change in the Si isotopic composition of silicified oceanic crusts (following their interaction with early Earth seawater) reworked as the granitoid melting sources. This trend was then carefully described as indicating a shift in the Si isotopic composition of the seawater towards heavier values. This change nicely parallels that observed in cherts and BIFs, which evolve rapidly from negative values in the Eoarchean ($-0.3 < \delta^{30}\text{Si} < -2.5$, André et al, EPSL 2006) to predominantly positive values in the Paleoproterozoic ($-0.5 < \delta^{30}\text{Si} < +2.5$, e.g. Sun et al, EPSL 2023). These two mimetic tendencies jointly support a seawater progressively enriched in ^{30}Si in less than 300Ma between 3.8 and 3.5 Ga. I find this part of the research innovative and well supported by excellent and timely data and therefore of great interest to the scientific community. I would be delighted to see this part of the study published soon.

According to the authors, the observed change is only possible because of a massive flux of heavy silicon from the weathering of land surfaces (33% of the total flux according to the authors) and they interpret their data as evidence of the intensification of continental weathering 3.7 billion years ago, which becomes the main claim of their contribution. This would overturn the prevalent idea that land masses only appeared later, in the Meso-Neoproterozoic period. However, I am afraid I cannot follow them in this assertion for two major problematic reasons, which lead me to propose that this interesting manuscript be acceptable only after a thorough revision.

My first major problem. If the ocean is well mixed every few thousand years since the beginning of the Earth (as proposed by Liu et al., *Geochem. Persp. Lett.* 31, 54-59, 2024), it implies maintaining silicon supersaturation throughout the ocean for a sufficiently long period to ensure the widespread silicification of the seafloors observed from the Eoarchean onwards (e.g. Bindeman & O'Neil, EPSL 2022). To constantly maintain a small degree of Si supersaturation at around 1 micromoles of Si within the whole Eo-Paleoproterozoic Ocean (taken as volumetrically similar to the modern one) requires supplying at least 1000 Tmol per year of silica to the global ocean, of which 333 Tmol would be extracted from land surfaces (according to the authors proposal of supposed 33% terrestrial input). This means 50 times more than is supplied today by continental inputs (6.1 Tmol, cf Tréguer et al., *Biogeosciences* 18, 1269-1289, 2021). I doubt this could be provided by early emergent land surfaces (Cf. below other comment 6).

My second major problem. The hydrothermal input of Si during the Archean must have been extremely important given the large heat flux of the early Earth and the alkalinity of hydrothermal fluids (Shibuya et al., 2010, *Precambrian Research* 182 230–238). It is therefore not enough to assert, without any proof as the authors do, that it is unlikely that hydrothermal discharge suddenly increased by 2 to 3 times between 3.8 Ga and 3.5 Ga to reject hydrothermal input as the main driver of

the changes observed in the Si isotopic evolution of seawater. On the contrary, I would argue that it's entirely plausible that hydrothermal activity plays a key role, and here's why. After the evaporation of the ocean by the most important impacts at the end of the period of heavy bombardment on Earth (around 4-3.9 Ga), the renewed ocean had plenty of time to generate (or regenerate) its Si saturation thanks to the silica continuously emitted by hydrothermal activities on the ridge. This would have taken around 6-7 million years, with an input of 1000Tmol per year (for a ridge discharge of 10^{16} Kg yr⁻¹ at 350°C with a Si content of 20mmol/l if no silica is precipitated during the mixture with the seawater), but probably a little longer (20-100 10^6 years) depending on the fraction of silica precipitated at a local level close to the ridge during the fluid mixtures. This could be the period during which no (or limited) massive silicification was recorded in the granitoid source (specimens in grey in Fig. 1A). As soon as silicon saturation was reached on a global scale, silica may have begun to precipitate throughout the world, suddenly separating the ²⁸Si (in opal) from the ³⁰Si in the residual ocean fluid, and this precipitation may then have been continuous due to the gradual cooling of the ocean. (as already proposed by Robert and Chaussidon in their 2006 pioneering work). This early silica precipitation would have developed from a basaltic reservoir (with $\delta^{30}\text{Si}=-0.29$) and would therefore have mostly negative Si isotopic compositions like those observed in the BIFs and cherts of the Isua Eoarchean (André et al., 2006). The progressive precipitation of silica by cooling seawater would then rapidly increase the ³⁰Si content in the global ocean, introducing a rapidly increasing positive signature in the silicified sources of the 3.8 to 2.9 Ga granitoids. The onus is on the authors to demonstrate that such a much simpler scenario cannot explain their data.

Other comments

1. Title and Lines 32, 33, 54, 68, 142, 160, 223, etc... referring to "Continental" weathering, sources, surfaces, crust, runoff, or silica. The adjective 'continental' is not at all appropriate here as it is reminiscent of a modern analog. What does 'continental' mean in the Eo-Paleoarchean when there is still much debate about the growth of the crust between 4 and 3Ga, the relative proportions of mafic and felsic rocks in the crust before 3.2 Ga and the origin of most early Earth felsic rocks? I recommend a more neutral formulation, such as 'emergent' sources, surfaces and weathering.
2. Lines 63-65 the 2 cited references (22,23) do not refer to the today conditions but the Precambrian ones. So please rephrase.
3. Lines 113-120: I'm wondering about the behavior of Ti isotopes during silicification of the seafloor. Are there any data available?
4. Lines 148-152 and figure 2. Honestly, it is tricky to determine whether the Si isotopic change observed among the granitoid signatures are gradual or abrupt due to the small amount of data still available between 3.5 and 3.7 Ga.
5. Lines 158-159 and Supplementary materials: "Mass balance models" The isotopic composition of the hydrothermal input $\delta^{30}\text{Si}_{\text{Hydro}}$ can be replaced by $\delta^{30}\text{Si}_{\text{basalt}}$. This is true if the hydrothermal Si vector is mostly controlled by the on-ridge high-temperature hydrothermal inputs. However, on modern Earth, low-temperature (<60°C) hydrothermal activity outside the ridges (off-ridge systems), may be as high as 2×10^{16} kg yr⁻¹, about two to three orders of magnitude times that circulating through the on-ridge hydrothermal systems (with a high temperature circulation of $\sim 4 \times 10^{12}$ kg yr⁻¹). During the Paleoarchean, such a low-T hydrothermal off-ridge system supplied the ocean with dissolved silicon bearing a very positive Si isotopic signature ($\delta^{30}\text{Si} \gg +0.4$) as attested by the precipitated quartz ($\delta^{30}\text{Si} = +0.4$, André et al., EPSL 2022) in all the veins developed during the late hydraulic fracturing of the silicified seafloor caps.
6. Lines 160-163: This discussion must be nuanced because weathering desilicifications are recognized as being very variable depending on the climate and rock types, with a much higher production of smectites (compared to kaolinite) in polar-temperate regions and on basaltic substrates (Bayon et al. GCA, 229, 147-161, 2018). As smectites do not fractionate silicon isotopes too much (Bayon et al. 2018), we do not expect a significant runoff of heavier Si (positive $\delta^{30}\text{Si}$) to the ocean if smectite is the dominant alteration product as would be the case if the Archean emerging crust is more mafic than felsic. In addition, recent models of the modern terrestrial Si cycle reveal that, due to incongruent weathering, much of the Si released during the weathering of silicate minerals (up to 94%, Rahman & Trower, 2023) remains stored in the terrestrial Si sinks, suggesting that surface weathering does not deliver too much heavy Si to the ocean. Therefore, if the very low sedimentation rates observed for mudstones around Archean greenstone belts (e.g. Kamber and Tomlinson, Chem. Geology 2019, 511, 123-151) confirms that only a small fraction of volcanic edifices were above sea level and subject to weathering, then the contribution of weathering to the delivery of a heavy silicon isotopic signature to the ocean would be extremely limited.

Reviewer #2 (Remarks to the Author):

This paper provides an interesting paper on the onset of continental weathering 3.6 Ga ago, based on a range of new geochemical data, mainly Si and Ti isotopes. The topic is of great interest to the Earth Sciences community and the paper is well written and illustrated. Based on these data, the authors propose that significant continental weathering occurred earlier than previously thought and quantify an increase in silicon continental erosion fluxes to the ocean of +33 +/- 7% between 3.8 and 3.6 Ga. Overall, the data and methods are clearly described, of very good quality, and convincing enough to demonstrate such an increase. However, the uncertainty in this flux should be clarified (or re-evaluated), as it might be underestimated given the relatively small change observed in the Si isotope data compared to the variability of the isotopic signatures in the different samples. See more on my only two concerns (moderate / minor) below.

1) Concerns about the significance of the 0.18 pmil shift.

I am not sure that the uncertainty has been fully propagated to the % change in Si flux from land to ocean, since the calculated uncertainties are small (33 +/- 7%, whereas more than 1/3 of the shift, 0.06 pmil, could be within the sample variability).

In this context, I do not understand the errors reported in lines 503-504. Using Table S1, we find an average of the capital delta = 0.18 +/- 0.05 pmil for TTG younger than 3.6 Ga and 0.04 +/- 0.05 (1SD) for samples older than 3.6 Ga, while the SD provided lines 504-504 are only 0.02 and 0.03 pmil. The shift is then 0.18-0.04 = 0.14 pmil, and 1 SD (0.05 pmil) could only explain > 33% of this shift... L 123-125 the averages are given for two more restricted age ranges, but still the SD should be 0.05 or 0.04 pmil based on Table S1. In the main text, the error is reported as SE, but as the number of samples is small (n <10), SD may be more appropriate than SEM.

It is necessary to clarify how the means and their SD were calculated and to identify the samples more clearly in Fig. 1 and/or Fig. 2, as the models are all based on the 0.18 pmil shift.

Fig. 2 In relation to the above comments, this Fig. 2 should show individual error bars from this study, at least on the d30Si capitals, and possibly also those from previous literature. As for Fig. S1, the legend of Fig. 2 should mention the difference between bold and plain line symbols.

2) Preferred scenario

L263-267 Some justification/references as to why the preferred scenario is when "the proportion of the continental Si input) is equivalent to the increase in the amount of Si added to the melting source of the granitoids." Even if this looks like a reasonable assumption, it would need some justification and what it implies: e.g. silica precipitation linearly correlated with Si concentration? No change in other parameters, ocean volume, hydrothermal inputs?....

Communications Earth & Environment is committed to improving transparency in authorship. As part of our efforts in this direction, we are now requesting that all authors identified as 'corresponding author' create and link their Open Researcher and Contributor Identifier (ORCID) with their account on the Manuscript Tracking System prior to acceptance. ORCID helps the scientific community achieve unambiguous attribution of all scholarly contributions. You can create and link your ORCID from the home page of the Manuscript Tracking System by clicking on 'Modify my Springer Nature account' and following the instructions in the link below. Please also inform all co-authors that they can add their ORCIDs to their accounts and that they must do so prior to acceptance.

Version 1:

Decision Letter:

Dear Professor Greber,

Your revised manuscript titled "Silicon isotopic signatures of granitoids support increased weathering of subaerial land around 3.7 billion years ago" has now been seen by original reviewer #2 and a new reviewer (#3), and we include their comments at the end of this message. They find your work of interest, but some important points are raised. We are interested in the possibility of publishing your study in Communications Earth & Environment, but would like to consider your responses to these concerns and assess a revised manuscript before we make a final decision on publication.

We therefore invite you to revise and resubmit your manuscript, along with a point-by-point response that takes into account the points raised. Please highlight all changes in the manuscript text file. When you revise your manuscript, please consider the following editorial thresholds as well:

**Address the discussion points from reviewer #1 regarding the scenario for surface weathering enhancement to explain changes in granitoid Si isotopic signatures and discuss alternative scenarios.

**Provide additional details and discussion on the formation of subaerial clays, as pointed out by reviewer #3.

Please submit your point-by-point responses as a separate file, distinct from your cover letter where you can add responses to the Editors' comments that you do not want to be made available to the reviewers. Word files are preferred. We

recommend that any figures, tables or graphs that are included in the response to reviewers are also included in the main article or Supplementary Information.

Please use the following link to submit your revised manuscript, point-by-point response to the referees' comments (which should be in a separate document to any cover letter), a tracked-changes version of the manuscript (as a PDF file) and the completed checklist:

Link Redacted

We hope to receive your revised paper within six weeks; please let us know if you aren't able to submit it within this time so that we can discuss how best to proceed. If we don't hear from you, and the revision process takes significantly longer, we may close your file. In this event, we will still be happy to reconsider your paper at a later date, as long as nothing similar has been accepted for publication at Communications Earth & Environment or published elsewhere in the meantime.

Please do not hesitate to contact us if you have any questions or would like to discuss these revisions further. We look forward to seeing the revised manuscript and thank you for the opportunity to review your work.

Best regards,

Mojtaba Fakhraee, PhD
Editorial Board Member
Communications Earth & Environment
orcid.org/0000-0002-2461-6374

Carolina Ortiz Guerrero, PhD
Associate Editor
Communications Earth & Environment

EDITORIAL POLICIES AND FORMATTING

Editorial Policy: [Policy requirements](https://www.nature.com/documents/nr-editorial-policy-checklist.pdf) (Download the link to your computer as a PDF.)

- Behavioural and social science
- Ecological, evolutionary & environmental sciences
- Life sciences

<https://www.nature.com/documents/nr-reporting-summary.zip>

Furthermore, please align your manuscript with our format requirements, which are summarized on the following checklist: [Communications Earth & Environment formatting checklist](https://www.nature.com/documents/commsj-phys-style-formatting-checklist-article.pdf)

and also in our style and formatting guide [Communications Earth & Environment formatting guide](https://www.nature.com/documents/commsj-phys-style-formatting-guide-accept.pdf) .

*** DATA: Communications Earth & Environment endorses the principles of the Enabling FAIR data project (<http://www.copdess.org/enabling-fair-data-project/>). We ask authors to make the data that support their conclusions available in permanent, publically accessible data repositories. (Please contact the editor if you are unable to make your data available).

All Communications Earth & Environment manuscripts must include a section titled "Data Availability" at the end of the Methods section or main text (if no Methods). More information on this policy, is available at <http://www.nature.com/authors/policies/data/data-availability-statements-data-citations.pdf>.

- Unique identifiers (such as DOIs and hyperlinks for datasets in public repositories)

- Accession codes where appropriate
- If applicable, a statement regarding data available with restrictions
- If a dataset has a Digital Object Identifier (DOI) as its unique identifier, we strongly encourage including this in the Reference list and citing the dataset in the Data Availability Statement.

If a community resource is unavailable, data can be submitted to generalist repositories such as [figshare](https://figshare.com/) or [Dryad Digital Repository](http://datadryad.org/). Please provide a unique identifier for the data (for example a DOI or a permanent URL) in the data availability statement, if possible. If the repository does not provide identifiers, we encourage authors to supply the search terms that will return the data. For data that have been obtained from publically available sources, please provide a URL and the specific data product name in the data availability statement. Data with a DOI should be further cited in the methods reference section.

REVIEWER COMMENTS:

Reviewer #1 (Remarks to the Author):

I would like to thank the authors for taking the time to carefully address my comments. Their detailed rebuttal highlights the immense challenges in visualizing early Earth conditions and in extrapolating from present-day evidence to those ancient environments. For instance, as the authors emphasize, current models of impact flux in the inner solar system—largely informed by lunar sample archives—still require significant refinement, particularly regarding the late heavy bombardment hypothesis and its impact on Earth. Similarly, our understanding of modern hydrothermal silicon fluxes to the ocean remains very limited, making it extremely difficult to quantify, because the Si balance linked to off-ridge hydrothermal fluxes, which are of the order of surface runoff, has yet to be assessed. Modern Si hydrothermal fluxes represent one of the most persistent poorly constrained inputs to the ocean's chemical balance today, rendering any extrapolation to Archean conditions highly speculative and fraught with uncertainty.

Having reviewed the revised manuscript, I remain skeptical of the proposed scenario for surface weathering enhancement to explain changes in granitoid Si isotopic signatures, mainly because (1) it does not adequately address the question of how dissolved silicon supersaturation of the vast global ocean volume could have been consistently maintained by surface weathering to silicify the seafloor during the Hadean-Archean transition - a question I identified as the main concern in my previous review ; (2) it models early Earth hydrothermal inputs by extrapolating global modern counterparts whose isotopic and silica signatures remain poorly assessed and require refinements.

Nevertheless, I acknowledge that the authors have made significant efforts to clarify several previously ambiguous or unclear points, particularly by (1) better specifying the role of clay formation in surface-to-ocean transfer processes following Archean weathering and its contribution to developing a heavier silicon isotopic signature under a possible mix of kaolinite-smectite clays, (2) proposing a more precise and nuanced title, namely "Silicon isotopic signatures of granitoids support increased weathering of subaerial land around 3.7 billion years ago". Additionally, I recognize that this study, underpinned by excellent new timely isotopic data, may contribute to a growing body of evidence suggesting the possible existence of some emergent land prior to 3.5 billion years ago. On this basis, I can recommend the manuscript for publication as it stands.

PS: "The title, which was correctly changed in the revised version with the red additions, seems to have been reverted to the original in the resubmitted manuscript, retaining the inappropriate term 'continental.'" Please check!

Reviewer #3 (Remarks to the Author):

As replacement for reviewer #2, I read the manuscript for the first time and I highly enjoyed it. I think, the findings and implications are very important to understand early Earth processes and will advance the field in assessing marine element budgets over geological time. I find the edits to the points raised by former reviewer #2 well-handled and incorporated in the revised manuscript. I have no further comments on this aspect.

Nevertheless, I have a number of minor comments (see below) to improve clarification. In addition, I have a major concern referring to the formation of subaerial clays, especially to the ambient conditions of formation, such as pH. Experimentally, it was shown that clays form in alkaline water, which stands in contrast to the model outcome the authors reference. I urge the authors to edit this section (see below for details) and to be a bit more careful about the implications of clay type and place of formation.

Taking these points into account, I'm in favor of publication.

Minor/moderate comments:

Line 42-44: a short explanation why lakes and continental shelves are important for the origin and evolution of life would be valuable.

Line 71: what about chert and BIF precipitation in addition to clay-rich sediments? Also the referenced publications, while acknowledging the authigenic clay-sink, concentrate mainly on Precambrian chert formation.

Line 83: I don't understand the reference to 200Ma here. You state before that the oldest modern oceanic crust is 200Ma and that it should be faster due to vertical tectonics in the Archean (at least I infer that from the 'oldest seafloor'; time clarification would be good here). I guess, the oldest Archean crust should be much younger than 200Ma?!

Line 95 – 97: small stylistic comment: you first write the age range from young to old (3.93 to 4.02Ga), then from old to young (3.80 to 2.94Ga). Consistency would be nice.

Line 107-108: $\Delta 30\text{Si}$ is also very often used as an equivalent to the fractionation factor ϵ . Might be helpful to add this here as well.

Line 125: it is not clear to me to which variable ($\delta 30\text{Si}$ or $\Delta 30\text{Si}$) the reported data is referring to?

Line 129: ..younger than 3.60 Ga, ...

Line 151: subaerial weathering means, that you weather/dissolve minerals, that are in general lighter in $\delta 30\text{Si}$ compared for example a fluid (e.g. seawater). Reverse weathering (precipitation) is the process that forms clays, that take up the light 28Si and then heavy 30Si signatures could be transported by a river to the oceans. As it is now in the manuscript, the statement is not correct.

Line 157: no explanation/definition is given what reverse weathering actually is.

Line 163: I find the formulation 'left behind' not very fitting here. The formation of clay minerals is a process of its own and not sth that is left behind by chem. weathering. Weathering 'leaves behind' dissolved cations, from which clays can precipitate.

Line 176ff: I find this section highly counterintuitive. My first impression was that you can not form clays in acidic (rain) waters. I do see that in the referenced modelling approach, clays form from weathering of basalt at a (rain water) pH of 5.5. However, to the best of my knowledge, this was never shown in any experiments and the only experiments conducted using Archean seawater conditions highlight the importance of an alkaline pH, with a threshold at 8.7 (see Tosca et al., 2011, 2016). Additionally, was there already enough Al available from the TTG suites to induce Al-bearing clay precipitation? In the absence of Al, Mg-Si clays precipitate with a- so far- unknown Si isotope fractionation factor. I find it hard to imagine, that you form 'enough' clays on these early continents, that it can already impact river and ocean waters. An alternative interpretation would be that the Al released by dissolving/weathering of TTG suites or even as particulate would be transported by rivers to the ocean and the clay formation would then occur within the early marine sediments, where ambient pH and saturation states are more likely to change, promoting clay formation. Additionally, a recent study by Geilert et al. (2024) shows that Si isotope signatures in clays are not stable until mature clay phases are reached. Therefore, I would be careful interpreting the light 30Si in shales as primary signature, as they could have been impacted by diagenesis. Summarizing, I would suggest to formulate your hypothesis of the occurrence of specific subaerial clays much more carefully and discuss as well the published experimental data, which are contradicting the model results.

Line 320: colloquial language

Communications Earth & Environment is committed to improving transparency in authorship. As part of our efforts in this direction, we are now requesting that all authors identified as 'corresponding author' create and link their Open Researcher and Contributor Identifier (ORCID) with their account on the Manuscript Tracking System prior to acceptance. ORCID helps the scientific community achieve unambiguous attribution of all scholarly contributions. You can create and link your ORCID from the home page of the Manuscript Tracking System by clicking on 'Modify my Springer Nature account' and following the instructions in the link below. Please also inform all co-authors that they can add their ORCIDs to their accounts and that they must do so prior to acceptance.

If you experience problems in linking your ORCID, please contact the Platform Support Helpdesk.

Version 2:

Decision Letter:

Dear Professor Greber,

Your revised manuscript titled "Silicon isotopic signatures of granitoids support increased weathering of subaerial land around 3.7 billion years ago" has now been seen by our original reviewer #3, whose comments appear below. In light of their advice we are delighted to say that we are happy, in principle, to publish a suitably revised version in Communications Earth & Environment.

We therefore invite you to revise your paper one last time to address the remaining concerns of our reviewers. At the same time we ask that you edit your manuscript to comply with our format requirements and to maximise the accessibility and therefore the impact of your work.

EDITORIAL REQUESTS:

****Please take care to match our formatting and policy requirements. We will check revised manuscript and return manuscripts that do not comply. Such requests will lead to delays. ****

SUBMISSION INFORMATION:

OPEN ACCESS:

Communications Earth & Environment is a fully open access journal. Articles are made freely accessible on publication. For further information about article processing charges, open access funding, and advice and support from Nature Research, please visit <https://www.nature.com/commsenv/open-access>

Link Redacted

Best regards,
Mojtaba Fakhraee, PhD
Editorial Board Member
Communications Earth & Environment
orcid.org/0000-0002-2461-6374

Carolina Ortiz Guerrero, Ph.D.
Associate Editor,
Communications Earth & Environment
Consulting Editor, Communications Sustainability

REVIEWERS' COMMENTS:

Reviewer #3 (Remarks to the Author):

The authors addressed my comments well and clarified clay formation in continental, acidic waters. Therefore, I am pleased with the edits and recommend the manuscript for publication.

Reviewer #1 (Remarks to the Author):

The authors demonstrate a change in the Si isotopic composition ($\delta^{30}\text{Si}$) of granitoids between 4.0 and 3.5 Ga from about -0.25 per mil to +0.1 per mil. This evolution is explained as reflecting a change in the Si isotopic composition of silicified oceanic crusts (following their interaction with early Earth seawater) reworked as the granitoid melting sources. This trend was then carefully described as indicating a shift in the Si isotopic composition of the seawater towards heavier values. This change nicely parallels that observed in cherts and BIFs, which evolve rapidly from negative values in the Eoarchean ($-0.3 < \delta^{30}\text{Si} < -2.5$, André et al., EPSL 2006) to predominantly positive values in the Paleoarchean ($-0.5 < \delta^{30}\text{Si} < +2.5$, e.g. Sun et al., EPSL 2023). These two mimetic tendencies jointly support a seawater progressively enriched in ^{30}Si in less than 300Ma between 3.8 and 3.5 Ga. I find this part of the research innovative and well supported by excellent and timely data and therefore of great interest to the scientific community. I would be delighted to see this part of the study published soon.

According to the authors, the observed change is only possible because of a massive flux of heavy silicon from the weathering of land surfaces (33% of the total flux according to the authors) and they interpret their data as evidence of the intensification of continental weathering 3.7 billion years ago, which becomes the main claim of their contribution. This would overturn the prevalent idea that land masses only appeared later, in the Meso-Neoproterozoic period. However, I am afraid I cannot follow them in this assertion for two major problematic reasons, which lead me to propose that this interesting manuscript be acceptable only after a thorough revision.

We thank the reviewer for having taken the time to comment on our manuscript and to provide a detailed review. We like to mention that there is already evidence for land emergence ≥ 3.5 Ga, including Sr isotopes from barytes (Roerdink et al., 2022), machine learning evidence that 20% of basalts erupted subaerially 3.8 Ga (Liu et al., 2024), and from the geological and sedimentary rock record in general, e.g. (Jacobsen and Dymek, 1988; Boyd et al., 2024). Therefore, our findings don't fully overturn the existing understanding but provide a more detailed perspective on how these emerged lands influenced element fluxes to the oceans.

My first major problem. If the ocean is well mixed every few thousand years since the beginning of the Earth (as proposed by Liu et al., *Geochem. Persp. Lett.* 31, 54-59, 2024), it implies maintaining silicon supersaturation throughout the ocean for a sufficiently long period to ensure the widespread silicification of the seafloors observed from the Eoarchean onwards (e.g. Bindeman & O'Neil, EPSL 2022). To constantly maintain a small degree of Si supersaturation at around 1 micromoles of Si within the whole Eo-Paleoarchean Ocean (taken as volumetrically similar to the modern one) requires supplying at least 1000 Tmol per year of silica to the global ocean, of which 333 Tmol would be extracted from land surfaces (according to the authors proposal of supposed 33% terrestrial input). This means 50 times more than is supplied today by continental inputs (6.1 Tmol, cf Tréguer et al., *Biogeosciences* 18, 1269-1289, 2021). I doubt this could be provided by early emergent land surfaces (Cf. below other comment 6).

My second major problem. The hydrothermal input of Si during the Archean must have been extremely important given the large heat flux of the early Earth and the alkalinity of hydrothermal fluids (Shibuya et al., 2010, *Precambrian Research* 182 230–238).

The arguments below have been added to the manuscript in a summarized form (see lines 281-300).

We are unsure how this 1000 Tmol were calculated, this value however seems rather high. Taking the values given by the reviewer, and assuming an absence of a Si flux from land, the 1000 Tmol Si would need to be provided by the hydrothermal input alone. Given that the modern hydrothermal Si flux is estimated to be between 0.4 Tmol (Frings et al., 2016; DeMaster, 2019) and 1.7 Tmol (Tréguer et al., 2021), a flux of 1000 Tmol would suggest an early Archean hydrothermal activity that was ~ 580 to ~ 1660 times that of the modern, if the Si concentration of the hydrothermal flux stayed similar. Even if the Si concentration in the hydrothermal flux was four times as high as today (maximum estimate by Shibuya et al., 2010), the required early Archean hydrothermal activity would be ~ 145 to ~ 415 times higher than in the modern.

To evaluate if this is likely, we calculate a conservative estimate for the hydrothermal Si flux in the early Archean. Isson and Planavsky, (2018) estimated that the hydrothermal Si flux 3.5 billion years ago was less than 5 times higher compared to the modern. Lowell and Keller, (2003) estimated based on heat loss modelling that the hydrothermal Fe flux 3.8 billion years ago is below 8.5 times that of the modern. Two points are to note; (i) Lowell and Keller, (2003) did not consider lower Fe concentrations in hydrothermal fluids for the early Archean as suggested by Shibuya et al. (2010), which makes their estimate applicable for our purpose, (ii) estimating the heat loss of the early Archean is challenging as the relationship that links mantle temperature with convective heat transport is unknown (discussed in Dauphas et al., 2024), resulting in estimates of a heat flux that did not change much since 4 billion years (Korenaga, 2008) to being around 3 times higher (Patočka et al., 2020). Thus, a priori,

it is uncertain that a higher heat flux existed in the early Archean, which makes the estimates of Lowell and Keller (2003) conservative for our calculations.

In any case, using the estimates from Isson and Planavsky (2018) as well as Lowell and Keller (2003), and given that the Si concentration in the hydrothermal flux might have been up to 4 times higher than today (Shibuya et al., 2010), the hydrothermal Si input 3.8 billion years ago could have been 5 to 34 times higher than present. This is much lower than what would be required to add 1000 Tmol Si per year solely by hydrothermal fluids to the ocean.

Furthermore, depending on which of the proposed modern Si cycles is used (e.g. Frings et al., 2016, DeMaster, 2019 or Tréguer et al., 2021), and considering a 5 to 34 times higher hydrothermal Si flux in the early Archean, our calculations of the land runoff after around 3.7 billion years ago indicate a continental supply that was 0.2 to 4.4 times the present levels, which is a plausible result.

It is therefore not enough to assert, without any proof as the authors do, that it is unlikely that hydrothermal discharge suddenly increased by 2 to 3 times between 3.8 Ga and 3.5 Ga to reject hydrothermal input as the main driver of the changes observed in the Si isotopic evolution of seawater. On the contrary, I would argue that it's entirely plausible that hydrothermal activity plays a key role, and here's why. After the evaporation of the ocean by the most important impacts at the end of the period of heavy bombardment on Earth (around 4-3.9 Ga), the renewed ocean had plenty of time to generate (or regenerate) its Si saturation thanks to the silica continuously emitted by hydrothermal activities on the ridge. This would have taken around 6-7 million years, with an input of 1000 Tmol per year (for a ridge discharge of 10^{16} Kg yr⁻¹ at 350°C with a Si content of 20 mmol/l if no silica is precipitated during the mixture with the seawater), but probably a little longer (20-100 10^6 years) depending on the fraction of silica precipitated at a local level close to the ridge during the fluid mixtures. This could be the period during which no (or limited) massive silicification was recorded in the granitoid source (specimens in grey in Fig. 1A). As soon as silicon saturation was reached on a global scale, silica may have begun to precipitate throughout the world, suddenly separating the ²⁸Si (in opal) from the ³⁰Si in the residual ocean fluid, and this precipitation may then have been continuous due to the gradual cooling of the ocean. (as already proposed by Robert and Chaussidon in their 2006 pioneering work). This early silica precipitation would have developed from a basaltic reservoir (with $\delta^{30}\text{Si} = -0.29$) and would therefore have mostly negative Si isotopic compositions like those observed in the BIFs and cherts of the Isua Eoarchean (André et al., 2006). The progressive precipitation of silica by cooling seawater would then rapidly increase the ³⁰Si content in the global ocean, introducing a rapidly increasing positive signature in the silicified sources of the 3.8 to 2.9 Ga granitoids. The onus is on the authors to demonstrate that such a much simpler scenario cannot explain their data.

While we appreciate the reviewer's perspective, we argue that this scenario is unlikely for several reasons. Firstly, based on all published estimates we are aware of, the hydrothermal flux to the oceans has remained constant or even decreased over time (as noted above). Secondly, there is significant evidence challenging the idea of a late heavy bombardment (e.g. Chapman et al., 2007; Boehnke and Harrison, 2016; Zellner, 2017; Mann, 2018; Mojzsis et al., 2019; Reimink et al., 2023), and with it the evaporation of the oceans at that time.

In summary, the hypothesis of a late heavy bombardment likely arises from a sampling bias in impact melts from the Moon, which skews the data against older samples, as well as misinterpretations of Ar-Ar age spectra. Additional evidence against a significant bombardment between 4.0 and 3.9 Ga comes from the lack of shock features in ancient (>3.7 Ga) zircon grains. Oceans have existed since >4.3 Ga (Lammer et al., 2018; Cameron et al., 2024), and recent literature argues against the existence of a terminal cataclysm 3.9 Ga. Therefore, in our view, the oceans have not evaporated and recondensed over the last 4.3 Ga.

In summary, the reviewer's hypothesis relies on the near-complete evaporation of the Earth's oceans as a result of asteroid bombardment, and based on recent evidence, both phenomena are seriously challenged.

In lines 142-144 we state now that in our view oceans existed since 4.3 Ga and that seawater was likely saturated in Si throughout the periods corresponding to the rocks studied in the manuscript.

Other comments

1. Title and Lines 32, 33, 54, 68, 142, 160, 223, etc... referring to "Continental" weathering, sources, surfaces, crust, runoff, or silica. The adjective 'continental' is not at all appropriate here as it is reminiscent of a modern analog. What does 'continental' mean in the Eo-Paleoarchean when there is still much debate about the growth of the crust between 4 and 3 Ga, the relative proportions of mafic and felsic rocks in the crust before 3.2 Ga and the origin of most early Earth felsic rocks? I recommend a more neutral formulation, such as 'emergent' sources, surfaces and weathering.

Changed according to the reviewer's suggestion to "land", "emerged crust" or another similar word.

2. Lines 63-65 the 2 cited references (22,23) do not refer to the today conditions but the Precambrian ones. So please rephrase.

Rephrased and additional references added for the modern situation.

3. Lines 113-120: I'm wondering about the behavior of Ti isotopes during silicification of the seafloor. Are there any data available?

We are not aware of available data. Almost pure chert ($\text{SiO}_2 > 95\text{wt}\%$) has TiO_2 concentrations of below 0.05 wt% (Abraham et al., 2011). Assuming conservatively that 10% chert has been mixed into the basaltic source (with $\sim 1\text{ wt}\%$ TiO_2) of Archean granitoids, the chert contributes only 0.55% of the total Ti to the system. Such a low amount of admixed Ti does not have a significant impact on the Ti isotopic composition (i.e. larger shift than 0.03‰), unless chert has a strongly fractionated isotopic composition (i.e. $\sim 5\%$), which is outside the range of Ti isotopic compositions observed as of yet. Thus, chert has likely no effect on the Ti isotopic compositions of the granitoids.

4. Lines 148-152 and figure 2. Honestly, it is tricky to determine whether the Si isotopic change observed among the granitoid signatures are gradual or abrupt due to the small amount of data still available between 3.5 and 3.7 Ga.

We changed "step-change" to "rapid increase".

5. Lines 158-159 and Supplementary materials: "Mass balance models" The isotopic composition of the hydrothermal input $\delta^{30}\text{Si}_{\text{Hydro}}$ can be replaced by $\delta^{30}\text{Si}_{\text{Basalt}}$. This is true if the hydrothermal Si vector is mostly controlled by the on-ridge high-temperature hydrothermal inputs. However, on modern Earth, low-temperature ($<60^\circ\text{C}$) hydrothermal activity outside the ridges (off-ridge systems), may be as high as $2 \times 10^{16} \text{ kg yr}^{-1}$, about two to three orders of magnitude times that circulating through the on-ridge hydrothermal systems (with a high temperature circulation of $\sim 4 \times 10^{12} \text{ kg yr}^{-1}$). During the Paleoarchean, such a low-T hydrothermal off-ridge system supplied the ocean with dissolved silicon bearing a very positive Si isotopic signature ($\delta^{30}\text{Si} > +0.4$) as attested by the precipitated quartz ($\delta^{30}\text{Si} = +0.4$, André et al., EPSL 2022) in all the veins developed during the late hydraulic fracturing of the silicified seafloor caps.

We interpret the explanation of the Si isotopic composition of the quartz precipitates in the fractures discussed by André et al., (2022) differently. In our view, André et al. (2022, see their Fig. 1) suggests that the $\delta^{30}\text{Si}$ value of +0.4‰ is the result of a mixture of isotopically heavy seawater and the release of isotopically light Si from underlying volcanic rocks. Therefore, we argue that these quartz veins do not indicate a new, isotopically distinct hydrothermal flux. Instead, the high $\delta^{30}\text{Si}$ observed in the quartz veins reflects the silicification of the seafloor by isotopically heavy seawater. Additionally, André et al. (2022) mentions that once an impermeable layer of chert is formed, the system becomes sealed. This implies that these fractures do not represent a long-lasting hydrothermal system.

For the model we now consider the composition of basalts and the hydrothermal input differently. The hydrothermal input (high and low temperature) is now defined as $-0.30 \pm 0.14\%$ as suggested by Rahman and Trower, (2023).

6. Lines 160-163: This discussion must be nuanced because weathering desilicifications are recognized as being very variable depending on the climate and rock types, with a much higher production of smectites (compared to kaolinite) in polar-temperate regions and on basaltic substrates (Bayon et al. GCA, 229, 147-161, 2018). As smectites do not fractionate silicon isotopes too much (Bayon et al. 2018), we do not expect a significant runoff of heavier Si (positive $\delta^{30}\text{Si}$) to the ocean if smectite is the dominant alteration product as would be the case if the Archean emerging crust is more mafic than felsic.

A possible impact of clay mineral formation and weathering processes on the Si isotope composition in the early Archean is discussed in lines 171-182.

We also like to state, that the emerged crust was likely mostly felsic in the early to mid-Archean, as it is more difficult to emerge a purely basaltic crust (Tang et al., 2024). Furthermore, the machine learning approach from Liu et al., (2024) states that in early Archean the "subaerial basaltic samples ... erupted on relatively stable emerged continental crust rather than short-lived oceanic islands". This speaks more for the existence of emerged land with a mixed lithological composition, and not exclusively mafic land.

Furthermore, following the <https://rruff.info/ima/> mineral evolution database, there seems to be more kaolinite than montmorillonite (smectite) in the Archean (record goes back to 3300 Ma). Overall, the ratio of the two clay minerals remains stable since 3300 Ma, but there is a lack of reported smectite occurrences between 3000 Ma and 2100 Ma (see Figure below). Thus, at least 3300 Ma ago kaolinite was produced during weathering, and there is no reason to assume that in the early Archean only smectite was produced during the chemical weathering of land.

Left: Total reported mineral occurrences of kaolinite and montmorillonite (binned in 100 Myrs) over time. Right: Ratio of kaolinite vs. montmorillonite over time. Overall, this ratio seems to remain at around 0.2 to 0.6. A ratio of zero indicates reported kaolinite, but no reports of montmorillonite (smectite) for this time period.

In addition, recent models of the modern terrestrial Si cycle reveal that, due to incongruent weathering, much of the Si released during the weathering of silicate minerals (up to 94%, Rahman & Trower, 2023) remains stored in the terrestrial Si sinks, suggesting that surface weathering does not deliver too much heavy Si to the ocean. Therefore, if the very low sedimentation rates observed for mudstones around Archean greenstone belts (e.g. Kamber and Tomlinson, *Chem. Geology* 2019, 511, 123-151) confirms that only a small fraction of volcanic edifices were above sea level and subject to weathering, then the contribution of weathering to the delivery of a heavy silicon isotopic signature to the ocean would be extremely limited.

It might be that there is significant incongruent weathering and a partial storage of Si in the continents in the modern. However, in average, the modelling of Rahman and Trower, (2023) still suggests a dissolved Si riverine runoff of ~6 Tmol per year with a $\delta^{30}\text{Si}$ of ~-1.21‰, i.e. ~20 Tm of Si is released during silicate mineral weathering and ~14 Tm are stored in terrestrial silica sinks.

We don't know if incongruent weathering in the Archean was equally abundant as today, but we do not see how this influences our argumentation. In contrast, if in the early Archean most dissolved Si was directly transported to the oceans without being stored in a terrestrial sink, then we would need to compare our calculations for the Archean to 20 Tm/year and not 6.2 Tm/year (see "first major problem"). Doing so would indicate that the Si runoff in the early Archean constituted only 0.05 to 1.6 times the modern Si runoff.

Reviewer #2 (Remarks to the Author):

This paper provides an interesting paper on the onset of continental weathering 3.6 Ga ago, based on a range of new geochemical data, mainly Si and Ti isotopes. The topic is of great interest to the Earth Sciences community and the paper is well written and illustrated. Based on these data, the authors propose that significant continental weathering occurred earlier than previously thought and quantify an increase in silicon continental erosion fluxes to the ocean of +33 +/- 7% between 3.8 and 3.6 Ga. Overall, the data and methods are clearly described, of very good quality, and convincing enough to demonstrate such an increase. However, the uncertainty in this flux should be clarified (or re-evaluated), as it might be underestimated given the relatively small change observed in the Si isotope data compared to the variability of the isotopic signatures in the different samples. See more on my only two concerns (moderate / minor) below.

We thank the reviewer for the comments and inputs and for having taken time to review our manuscript.

1) Concerns about the significance of the 0.18 pmil shift.

I am not sure that the uncertainty has been fully propagated to the % change in Si flux from land to ocean, since the calculated uncertainties are small (33 +/- 7%, whereas more than 1/3 of the shift, 0.06 pmil, could be within the sample variability).

In this context, I do not understand the errors reported in lines 503-504. Using Table S1, we find an average of the capital delta = 0.18 +/- 0.05 pmil for TTG younger than 3.6 Ga and 0.04 +/- 0.05 (1SD) for samples older than 3.6 Ga, while the SD provided lines 504-504 are only 0.02 and 0.03 pmil. The shift is then 0.18-0.04 = 0.14 pmil, and 1 SD (0.05 pmil) could only explain > 33% of this shift.... L 123-125 the averages are given for two more restricted age ranges, but still the SD should be 0.05 or 0.04 pmil based on Table S1. In the main text, the error is reported as SE, but as the number of samples is small (n <10), SD may be more appropriate than SEM.

We are sorry for the misunderstanding; this is now better indicated in the manuscript. The data in line 504 were given as standard errors. Part of the different interpretation of the reviewer and of us derives from the used dataset, the first only produced by us, and the second that also includes literature data. However, independent on which dataset is used, the student's t-test p-value comparing rocks with ages from 3.82 to 3.6 Ga and those younger than 3.6 Ga is always < 0.0001, showing that their $\Delta^{30}\text{Si}$ values are significantly different.

In the revised manuscript, for all of the modelling we use all the available data, ours and the already published one. This results in:

– $\Delta^{30}\text{Si}$ of granitoids with ages between 3.82 to 3.60 Ga = 0.075 ± 0.033 (2SE, n=12)
– $\Delta^{30}\text{Si}$ of granitoids with ages younger than 3.60 Ga = 0.160 ± 0.018 (2SE, n=67).

As their sample numbers are above 10, we argue that using the standard error as uncertainty for the model is justified. The errors of the Monte Carlo model might have been small as they were displayed at the 1-sigma level. We now report in the paper the 2-sigma level, yielding a result for the preferred scenario of 32 ± 15% Si contribution from the erosion of land.

It is necessary to clarify how the means and their SD were calculated and to identify the samples more clearly in Fig. 1 and/or Fig. 2, as the models are all based on the 0.18 pmil shift.

As it does not change the result, we simplified the chosen data for the model. It is now all data (ours and that published) except those from the dry magmatic differentiation series from Idiwahaa, i.e. all data displayed in Figure 2. The first set is all data with ages older than 3.6 Ga (n=12), and the second set is all data with ages younger than 3.6 Ga (n=58). We now state in the supplementary text of the manuscript (lines 18-120) that “the $\Delta^{30}\text{Si}$ value of granitoids younger than 3.60 Ga (i.e. +0.164 ± 0.009‰; 1SE, n= 58)” and “the $\Delta^{30}\text{Si}$ value of granitoid between 3.82 Ga to 3.60 Ga (i.e. +0.075 ± 0.015‰; 1SE, n=12)” are used, including the identification of the type of error (1SE) and the number of samples. We hope this helps in making the modeling more comprehensible.

Fig. 2 In relation to the above comments, this Fig. 2 should show individual error bars from this study, at least on the $\delta^{30}\text{Si}$ capitals, and possibly also those from previous literature. As for Fig. S1, the legend of Fig. 2 should mention the difference between bold and plain line symbols.

This has been corrected/modified according to the reviewer's suggestion. Errors were also added to Table S2.

2) Preferred scenario

L263-267 Some justification/references as to why the preferred scenario is when “the proportion of the continental Si input) is equivalent to the increase in the amount of Si added to the melting source of the granitoids.” Even if this looks like a reasonable assumption, it would need some justification and what it implies: e.g. silica precipitation linearly correlated with Si concentration? No change in other parameters, ocean volume, hydrothermal inputs?....

We added additional justification. The reason is as suspected by the reviewer: The Si that is additionally contributed to the seawater by the erosion of land needs to be compensated by a larger Si output (precipitation). We assume that this additional output is linearly correlated with the increase of Si added to the melting source of the granitoids.

References

- Abraham K., Hofmann A., Foley S. F., Cardinal D., Harris C., Barth M. G. and Andre L. (2011) Coupled silicon-oxygen isotope fractionation traces Archaean silicification. *Earth Planet. Sci. Lett.* **301**, 222–230. Available at: <http://dx.doi.org/10.1016/j.epsl.2010.11.002>.
- André L., Monin L. and Hofmann A. (2022) The origin of early continental crust: New clues from coupling Ge/Si ratios with silicon isotopes. *Earth Planet. Sci. Lett.* **582**, 117415. Available at: <https://doi.org/10.1016/j.epsl.2022.117415>.

- Boehnke P. and Harrison T. M. (2016) Illusory late heavy bombardments. *Proc. Natl. Acad. Sci. U. S. A.* **113**, 10802–10806.
- Boyd A. J., Rosing M. T., Harding M. A. R., Canfield D. E. and Hassenkam T. (2024) 3.7 billion year old detrital sediments in Greenland are consistent with active plate tectonics in the Eoarchean. *Commun. Earth Environ.* **5**, 201. Available at: <https://www.nature.com/articles/s43247-024-01376-w>.
- Cameron E. M., Blum T. B., Cavosie A. J., Kitajima K., Nasdala L., Orland I. J., Bonamici C. E. and Valley J. W. (2024) Evidence for oceans pre-4300 Ma confirmed by preserved igneous compositions in Hadean zircon. *Am. Mineral.* **109**, 1670–1681. Available at: <https://pubs.geoscienceworld.org/ammin/article/109/10/1670/637118/Evidence-for-oceans-pre-4300-Ma-confirmed-by>.
- Chapman C. R., Cohen B. A. and Grinspoon D. H. (2007) What are the real constraints on the existence and magnitude of the late heavy bombardment? *Icarus* **189**, 233–245.
- Dauphas N., Heard A. W., Rego E. S., Rouxel O., Marin-Carbonne J., Pasquier V., Bekker A. and Rowley D. (2024) Past and present dynamics of the iron biogeochemical cycle. *Ref. Modul. Earth Syst. Environ. Sci.*
- DeMaster D. J. (2019) The Global Marine Silica Budget: Sources and Sinks. In *Encyclopedia of Ocean Sciences* Elsevier. pp. 473–483. Available at: <http://dx.doi.org/10.1016/B978-0-12-409548-9.10799-7>.
- Frings P. J., Clymans W., Fontorbe G., De La Rocha C. L. and Conley D. J. (2016) The continental Si cycle and its impact on the ocean Si isotope budget. *Chem. Geol.* **425**, 12–36. Available at: <http://dx.doi.org/10.1016/j.chemgeo.2016.01.020>.
- Isson T. T. and Planavsky N. J. (2018) Reverse weathering as a long-term stabilizer of marine pH and planetary climate. *Nature* **560**, 471–475. Available at: <http://dx.doi.org/10.1038/s41586-018-0408-4>.
- Jacobsen S. B. and Dymek R. F. (1988) Nd and Sr isotope systematics of clastic metasediments from Isua, West Greenland: identification of pre-3.8Ga differentiated crustal components. *J. Geophys. Res.* **93**, 338–354.
- Korenaga J. (2008) Urey ratio and the structure and evolution of Earth's mantle. *Rev. Geophys.* **46**, 1–32.
- Lammer H., Zerkle A. L., Gebauer S., Tosi N., Noack L., Scherf M., Manuel E. P., Lee J., Mareike G. and Athanasia G. (2018) Origin and evolution of the atmospheres of early Venus, Earth and Mars. *Astron. Astrophys. Rev.* **26**, 1–72. Available at: <https://doi.org/10.1007/s00159-018-0108-y>.
- Liu C. T., Liu X. M. and Zhang Zhou J. (2024) Data-Driven Investigation Reveals Subaerial Proportion of Basalts Since the Early Archean. *Geophys. Res. Lett.* **51**.
- Lowell R. P. and Keller S. M. (2003) High-temperature seafloor hydrothermal circulation over geologic time and archean banded iron formations. *Geophys. Res. Lett.* **30**, 1–4.
- Mann A. (2018) Bashing holes in the tale of Earth's troubled youth. *Nature* **553**, 393–395. Available at: <https://www.nature.com/articles/d41586-018-01074-6>.
- Mojzsis S. J., Brasser R., Kelly N. M., Abramov O. and Werner S. C. (2019) Onset of Giant Planet Migration before 4480 Million Years Ago. *Astrophys. J.* **881**, 44. Available at: <http://dx.doi.org/10.3847/1538-4357/ab2c03>.
- Patočka V., Šrámek O. and Tosi N. (2020) Minimum heat flow from the core and thermal evolution of the Earth. *Phys. Earth Planet. Inter.* **305**, 106457. Available at: <https://doi.org/10.1016/j.pepi.2020.106457>.
- Rahman S. and Trower E. J. (2023) Probing surface Earth reactive silica cycling using stable Si isotopes: Mass balance, fluxes, and deep time implications. *Sci. Adv.* **9**, 1–13.
- Reimink J., Crow C., Moser D., Jacobsen B., Bauer A. and Chacko T. (2023) Quantifying the effect of late bombardment on terrestrial zircons. *Earth Planet. Sci. Lett.* **604**, 118007. Available at: <https://doi.org/10.1016/j.epsl.2023.118007>.
- Roerdink D. L., Ronen Y., Strauss H. and Mason P. R. D. (2022) Emergence of felsic crust and subaerial weathering recorded in Palaeoarchaeon barite. *Nat. Geosci.* **15**, 227–232. Available at: <https://www.nature.com/articles/s41561-022-00902-9>.
- Shibuya T., Komiya T., Nakamura K., Takai K. and Maruyama S. (2010) Highly alkaline, high-temperature hydrothermal fluids in the early Archean ocean. *Precambrian Res.* **182**, 230–238. Available at: <http://dx.doi.org/10.1016/j.precamres.2010.08.011>.
- Tang M., Chen H., Lee C.-T. A. and Cao W. (2024) Subaerial crust emergence hindered by phase-

driven lower crust densification on early Earth. *Sci. Adv* **10**, 1952. Available at:
<https://www.science.org>.

Tréguer P. J., Sutton J. N., Brzezinski M., Charette M. A., Devries T., Dutkiewicz S., Ehlert C.,
Hawkings J., Leynaert A., Liu S. M., Monferrer N. L., López-Acosta M., Maldonado M., Rahman
S., Ran L. and Rouxel O. (2021) Reviews and syntheses: The biogeochemical cycle of silicon in
the modern ocean. *Biogeosciences* **18**, 1269–1289.

Zellner N. E. B. (2017) Cataclysm No More: New Views on the Timing and Delivery of Lunar
Impactors. *Orig. Life Evol. Biosph.* **47**, 261–280. Available at:
<http://link.springer.com/10.1007/s11084-017-9536-3>.

Reviewer #1 (Remarks to the Author):

I would like to thank the authors for taking the time to carefully address my comments.

Their detailed rebuttal highlights the immense challenges in visualizing early Earth conditions and in extrapolating from present-day evidence to those ancient environments. For instance, as the authors emphasize, current models of impact flux in the inner solar system—largely informed by lunar sample archives—still require significant refinement, particularly regarding the late heavy bombardment hypothesis and its impact on Earth. Similarly, our understanding of modern hydrothermal silicon fluxes to the ocean remains very limited, making it extremely difficult to quantify, because the Si balance linked to off-ridge hydrothermal fluxes, which are of the order of surface runoff, has yet to be assessed. Modern Si hydrothermal fluxes represent one of the most persistent poorly constrained inputs to the ocean's chemical balance today, rendering any extrapolation to Archean conditions highly speculative and fraught with uncertainty.

Having reviewed the revised manuscript, I remain skeptical of the proposed scenario for surface weathering enhancement to explain changes in granitoid Si isotopic signatures, mainly because (1) it does not adequately address the question of how dissolved silicon supersaturation of the vast global ocean volume could have been consistently maintained by surface weathering to silicify the seafloor during the Hadean-Archean transition - a question I identified as the main concern in my previous review ; (2) it models early Earth hydrothermal inputs by extrapolating global modern counterparts whose isotopic and silica signatures remain poorly assessed and require refinements.

Nevertheless, I acknowledge that the authors have made significant efforts to clarify several previously ambiguous or unclear points, particularly by (1) better specifying the role of clay formation in surface-to-ocean transfer processes following Archean weathering and its contribution to developing a heavier silicon isotopic signature under a possible mix of kaolinite-smectite clays, (2) proposing a more precise and nuanced title, namely "Silicon isotopic signatures of granitoids support increased weathering of subaerial land around 3.7 billion years ago". Additionally, I recognize that this study, underpinned by excellent new timely isotopic data, may contribute to a growing body of evidence suggesting the possible existence of some emergent land prior to 3.5 billion years ago. On this basis, I can recommend the manuscript for publication as it stands.

We like to thank the reviewer again for having given us helpful insights that improved the manuscript.

Regarding point 1 of the Reviewer: we absolutely agree that hydrothermalism played a large role in maintaining the Si-saturation of Precambrian oceans, and indeed the implication of our models is that ~2/3 of oceanic Si in the Archean is required to be hydrothermally sourced.

Furthermore, it is widely accepted that due to the lack of an efficient biomediated sink for silica in the Archean, it has been suggested that the oceans were nearly saturated with silica, leading to the primary chemical precipitation of amorphous silica phases (e.g. Siever, 1992; Stefurak et al., 2015). Geological evidence supporting this idea includes the widespread presence of chert and diagenetically, silicified rocks, in Archean sequences (van den Boorn et al., 2007; Abraham et al., 2011; Stefurak et al., 2015).

We have added a number of sentences throughout the work to reiterate that we are basing our modelling on the view that Archean oceans remained Si-saturated – and that even with an increase in continental runoff, hydrothermalism still dominated the Si flux to the oceans at this time.

PS: "The title, which was correctly changed in the revised version with the red additions, seems to have been reverted to the original in the resubmitted manuscript, retaining the inappropriate term 'continental.'" Please check!

The title is now everywhere accurate.

Reviewer #3 (Remarks to the Author):

As replacement for reviewer #2, I read the manuscript for the first time and I highly enjoyed it. I think, the findings and implications are very important to understand early Earth processes and will advance the field in assessing marine element budgets over geological time. I find the edits to the points raised by former reviewer #2 well-handled and incorporated in the revised manuscript. I have no further comments on this aspect.

Nevertheless, I have a number of minor comments (see below) to improve clarification. In addition, I have a major concern referring to the formation of subaerial clays, especially to the ambient conditions of formation, such as pH. Experimentally, it was shown that clays form in alkaline water, which stands in contrast to the model outcome the authors reference. I urge the authors to edit this section (see below for details) and to be a bit more careful about the implications of clay type and place of formation.

Taking these points into account, I'm in favor of publication.

We thank the reviewer for having taken time to read and correct our manuscript. His/her input have strengthened our arguments and improved the quality of the manuscript overall. With regard to the broad point about clay mineral formation – we are puzzled to the authors suggestions here, and wonder if this is a confusion in terminology between our use of the term “weathering” and that of the Reviewers. We expand upon these aspects later.

Minor/moderate comments:

Line 42-44: a short explanation why lakes and continental shelves are important for the origin and evolution of life would be valuable.

Has been added

Line 71: what about chert and BIF precipitation in addition to clay-rich sediments? Also the referenced publications, while acknowledging the authigenic clay-sink, concentrate mainly on Precambrian chert formation.

In this work, we assume that the main marine authigenic sediments are BIFs and chert in the Precambrian; this is what we mean by “silica added to the seafloor in the Precambrian” – as this is the key lithology that is transferring the isotopically heavy oceanic Si to the source of the Archaean granitoids. We recognized that this sentence was not well crafted and thus modified it to makes this clearer (see line 71-77).

Line 83: I don't understand the reference to 200Ma here. You state before that the oldest modern oceanic crust is 200Ma and that it should be faster due to vertical tectonics in the Archean (at least I infer that from the ‘oldest seafloor’; time clarification would be good here). I guess, the oldest Archean crust should be much younger than 200Ma?!

It is rewritten and hopefully clarified (see lines 82-93). Here we are attempting to infer that, independent of the tectonic regime, silicifying and then burying oceanic crust to the melting depth could take a maximum of 200 Ma.

Line 95 – 97: small stylistic comment: you first write the age range from young to old (3.93 to 4.02Ga), then from old to young (3.80 to 2.94Ga). Consistency would be nice.

Changed

Line 107-108: $\Delta 30\text{Si}$ is also very often used as an equivalent to the fractionation factor ϵ . Might be helpful to add this here as well.

We have added this clarification (see lines 115-118).

Line 125: it is not clear to me to which variable ($\delta 30\text{Si}$ or $\Delta 30\text{Si}$) the reported data is referring to?

We have rephrased the sentence (see lines 134-144).

Line 129: ..younger than 3.60 Ga, ...

Corrected

Line 151: subaerial weathering means, that you weather/dissolve minerals, that are in general lighter in $\delta 30\text{Si}$ compared for example a fluid (e.g. seawater). Reverse weathering (precipitation) is the process that forms clays, that take up the light 28Si and then heavy $d30\text{Si}$ signatures could be transported by a river to the oceans. As it is now in the manuscript, the statement is not correct.

Throughout the manuscript we are using the term “chemical weathering” or “weathering” to describe the overall process of breakdown of primary minerals on land and reprecipitation of the ions into new secondary clay

minerals. We believe that the Reviewer, in their statement, is describing the processes involved in Marine Silicate Alteration (MAS), which is defined as a combination of “forward weathering” or “weathering” which is the breakdown of primary silicates, and “reverse weathering” which is the formation of secondary silicates – e.g. Geilert et al., (2023) or Geilert et al., (2024). This more specific usage is only concerned with the specific process of MAS via interaction of marine pore waters and terrigenous and biogenic marine sediments.

For instance, Opfergelt and Delmelle (2012) in their review paper on Si isotopes and formation of secondary clay minerals during terrestrial weathering, state:

*“In general, the Si isotopic composition of clay minerals formed during **weathering** is lighter ($\delta^{30}\text{Si} = -2.95$ to -0.16‰) than that of the parent silicate material (Fig. 3; Bern et al., 2010, Cornelis et al., 2010, Douthitt, 1982, Georg et al., 2007a, Georg et al., 2009a, Georg et al., 2009b, Opfergelt et al., 2010, Steinhöfel et al., 2011, Ziegler et al., 2005a, Ziegler et al., 2005b). This was shown to be the result of preferential incorporation of the light Si isotopes during mineral formation (Georg et al., 2007a, Ziegler et al., 2005a, Ziegler et al., 2005b). An equilibrium isotopic fractionation of $30\epsilon = -1.6\text{‰}$ was estimated from theoretical calculations between kaolinite and quartz (Méheut et al., 2007). This value matches that derived from field based-measurements of silicate weathering in Iceland ($30\epsilon = -1.5\text{‰}$; Georg et al., 2007a).”*

We prefer to use the more commonly utilized terminology, as the term “reverse weathering” is widely used to describe weathering reactions occurring in the ocean (during MAS, see above), i.e. the formation of clay minerals through the consumption of soluble cations, which releases acidity (CO_2), and is thus the opposite of terrestrial weathering reactions (e.g. Mackenzie and Kump, 1995; Isson and Planavsky, 2018). See for example also Rahman and Trower, (2023): *“In 1966, reverse weathering reactions were formally defined as a suite of reactions which had to occur in the ocean to maintain the mass balance of alkalinity, atmospheric CO_2 , and dissolved Si in the ocean (27).”*

Line 157: no explanation/definition is given what reverse weathering actually is.
See comment above, and the explanation now given in lines 166-169.

Line 163: I find the formulation 'left behind' not very fitting here. The formation of clay minerals is a process of its own and not sth that is left behind by chem. weathering. Weathering 'leaves behind' dissolved cations, from which clays can precipitate.
Left behind has been deleted

Line 176ff: I find this section highly counterintuitive. My first impression was that you can not form clays in acidic (rain) waters. I do see that in the referenced modelling approach, clays form from weathering of basalt at a (rain water) pH of 5.5. However, to the best of my knowledge, this was never shown in any experiments and the only experiments conducted using Archean seawater conditions highlight the importance of an alkaline pH, with a threshold at 8.7 (see Tosca et al., 2011, 2016).

Again, this seems to be a clash between the fields of marine silicate alteration, which generates and utilizes a high pH, high alkalinity environment, and terrestrial weathering which can occur at circumneutral to acidic conditions (indeed, carbonic acid is an important reactant in many overall weathering reactions). In modern conditions, kaolinite (a clay mineral known to strongly fractionate Si isotopes, Opfergelt and Delmelle 2012) is formed by chemical weathering of for example plagioclase and biotite (Nesbitt and Young, 1989; Jolicoeur et al., 2000), or through the transformation of initially produced smectite and montmorillonite to kaolinite (Altschuler et al., 1963; Karathanasis and Hajek, 1983), on land and under at least initially acidic conditions. For example, Karathanasis and Hajek, (1983) studied the transformation of smectite to kaolinite in naturally acid soil systems with pH values of 3.8 to 5.0.

Many other clay minerals that fractionate Si isotopes to different degrees like illite, smectite, and vermiculite are formed during the chemical weathering of emerged crust which happens under mildly acidic conditions as well (Nesbitt and Young, 1989; Opfergelt et al., 2012).

Both studies cited by the reviewer (i.e. Tosca et al., 2011, 2016) discuss the processes that dissolved material undergoes in the oceans. Tosca et al., (2016) suggests that greenalite precipitates from ocean water at a pH between 7.7 to 8.3. Tosca et al., (2011) discusses the conditions on how to form early diagenetic talc in marine carbonate platforms. These processes are representative of reverse weathering during marine silicate alteration, and not, we suggest on emerged land, also not for the early Archean.

Additionally, was there already enough Al available from the TTG suites to induce Al-bearing clay precipitation? In the absence of Al, Mg-Si clays precipitate with a- so far- unknown Si isotope fractionation factor.

The precise lithological composition of early land (~3.5 Ga) is disputed, but many studies argue that it contained above >50% felsic rocks (see Greber et al., 2017; Greber and Dauphas, 2019; Keller and Harrison, 2020; Ptáček et al., 2020; Garçon, 2021; Lipp et al., 2021; Roerdink et al., 2022). Furthermore, modelling shows an emerged crust in the early to mid-Archean was likely mostly felsic, as it is more difficult to emerge a purely basaltic crust (Tang et al., 2024). A lack of Al is also unlikely, as Al₂O₃ concentrations in basalts (~15 wt%) and granitoids/TTGs (~14.5 wt%) are similar (see e.g. Table S4 in Greber et al., 2017).

I find it hard to imagine, that you form 'enough' clays on these early continents, that it can already impact river and ocean waters.

Indeed, once land emerges, it is highly likely that its chemical alteration and erosion impacts the composition of ocean waters. The reason is that chemical weathering of land is a much more efficient sink for CO₂ than seafloor weathering (e.g. Krissansen-Totton et al., 2018). It is most likely that the total amount of chemical weathering reactions is largely independent of the areal extent of land, as it is defined by the required scavenging of CO₂ from the atmosphere. See for example Korenaga et al., (2017) that states: "*With a constant CO₂ input the ocean-atmosphere system and with all other climate variables held constant, varying the extent of continental exposure will result in disparate globally averaged weathering intensities and different steady-state atmospheric CO₂ concentrations. With less continental exposure, more intensive silicate weathering per unit area is required to balance a given carbon dioxide input flux*".

Another example that the areal extent of land is not correlated with the total amount of chemical weathering that takes place on continents can be found in the modern world. Nowadays, chemical weathering and CO₂ scavenging is strongly localized, and less than 9% of land is responsible for over 50% of the CO₂ sink (Hartmann et al., 2009). This highlights that the areal extent of land is secondary for the total amount of chemical weathering that takes place, and thus also clay formation.

Fig. 2. Global distribution of CO₂-consumption by chemical weathering. Note the global average for exorheic areas is ~2 t C km⁻² a⁻¹.

Figure from Hartmann et al., (2009), illustrating hotspots for CO₂ drawdown. Only 9% of the exorheic area is responsible for 50% of the CO₂ drawdown. It is to note that the regions with the highest CO₂ drawdown are also those with the highest flux of dissolved Si to the oceans (see Dürr et al., 2011).

An alternative interpretation would be that the Al released by dissolving/weathering of TTG suites or even as particulate would be transported by rivers to the ocean and the clay formation would then occur within the early marine sediments, where ambient pH and saturation states are more likely to change, promoting clay formation. We are unsure about the precise mechanism the reviewer suggests. Taking the three processes displayed in Figure 1 of Geilert et al., (2023) as a guide (i.e. marine silicate weathering, reverse weathering, coupled weathering), only reverse weathering can account for a significant increase in heavy Si isotopes in seawater. However, in the Archean, the absence of biogenic Si fixation led to a high Si concentration in seawater (nearly

saturated), making reverse weathering in any case a widespread phenomenon throughout the Precambrian (e.g. Isson and Planavsky, 2018).

Although we agree that additional cations provided by river runoff may contribute to clay formation in marine sediments, their overall impact is questionable, as a large number of cations that also originate from a non-continental source are present in the Archean ocean, such as Fe^{2+} , which can also be used for reverse weathering reactions (Isson and Planavsky, 2018).

Since the process suggested by the reviewer requires emerged lands and river runoff, and given the evidence outlined above that chemical weathering of these lands lead to the formation of clay minerals hosting light Si isotopes, we conclude that isotopically heavy river runoff was likely the most significant source of heavy Si isotopes to seawater from 3.8 to 3.6 Ga on.

We added these explanations on lines 160-177.

Additionally, a recent study by Geilert et al. (2024) shows that Si isotope signatures in clays are not stable until mature clay phases are reached. Therefore, I would be careful interpreting the light $\delta^{30}\text{Si}$ in shales as primary signature, as they could have been impacted by diagenesis.

The different components that are required to explain the $\delta^{30}\text{Si}$ in the cited shales are discussed in Savage et al., (2013) and include terrigenous detrital clay. Furthermore, the $\delta^{30}\text{Si}$ vs Al/Si correlation of the shales is opposite to that described in Geilert et al., (2024) for authigenic marine clay formation, making it unlikely that the shale's $\delta^{30}\text{Si}$ is dominated by authigenic marine clay minerals. We modified the sentence in lines 197-199 to clarify our argument regarding the Si isotopic composition of shales.

Summarizing, I would suggest to formulate your hypothesis of the occurrence of specific subaerial clays much more carefully and discuss as well the published experimental data, which are contradicting the model results. We like to mention that, in our view, assumptions and possible limitations on this matter were discussed in the previous version of the manuscript, particularly in lines 204-207, 217-218, and 491-496.

Many factors influence the Si isotope fractionation during chemical weathering of land and thus the $\delta^{30}\text{Si}$ value of river water, including the type of clay minerals that are initially formed, their potential further transformation during sustained alteration (e.g. montmorillonite to kaolinite), if the weathering reactions occur under kinetic or equilibrium conditions and whether or not Si gets adsorb on Fe-hydroxides. But altogether, these chemical weathering reactions occur on modern continents and under acidic conditions, resulting in river water transporting a heavy Si isotopic signature. Given our current understanding of the mid- to early-Archean, and the further explanations provided above, there is no reason to assume that several of the above processes that lead to an enrichment of heavy Si isotopes in water did not occur, such as adsorption of light isotopes on Fe-hydroxides and the production of isotopically light clay minerals during chemical weathering, either formed during fast kinetic chemical reactions, or alteration of e.g. feldspar or micas resulting in the formation of isotopically light kaolinite, for example.

The published experimental data mentioned by the reviewer above do not contradict these theories, as they investigate under which conditions oceanic pore waters interact with marine sediments, which is, in our view, not an analogue for the processes that occur during weathering of continental crust.

Finally, in an attempt to clarify our theories, we have extended this discussion as suggested by the reviewer (see lines 188-211).

Line 320: colloquial language

We changed "at face value" to "as is".

References

- Abraham K., Hofmann A., Foley S. F., Cardinal D., Harris C., Barth M. G. and Andre L. (2011) Coupled silicon-oxygen isotope fractionation traces Archaean silicification. *Earth Planet. Sci. Lett.* **301**, 222–230. Available at: <http://dx.doi.org/10.1016/j.epsl.2010.11.002>.
- Altschuler Z. S., Dwornik E. J. and Kramer H. (1963) Transformation of Montmorillonite to Kaolinite during Weathering. *Science (80-.)*. **141**, 148–152.

- van den Boorn S. H. J. M., van Bergen M. J., Nijman W. and Vroon P. Z. (2007) Dual role of seawater and hydrothermal fluids in Early Archean chert formation: Evidence from silicon isotopes. *Geology* **35**, 939–942.
- Dürr H. H., Meybeck M., Hartmann J., Laruelle G. G. and Roubex V. (2011) Global spatial distribution of natural riverine silica inputs to the coastal zone. *Biogeosciences* **8**, 597–620.
- Garçon M. (2021) Episodic growth of felsic continents in the past 3.7 Ga. *Sci. Adv.* **7**.
- Geilert S., Frick D. A., Abbott A. N. and Löhr S. C. (2024) Marine clay maturation induces systematic silicon isotope decrease in authigenic clays and pore fluids. *Commun. Earth Environ.* **5**, 3–10. Available at: <http://dx.doi.org/10.1038/s43247-024-01746-4>.
- Geilert S., Frick D. A., Garbe-Schönberg D., Scholz F., Sommer S., Grasse P., Vogt C. and Dale A. W. (2023) Coastal El Niño triggers rapid marine silicate alteration on the seafloor. *Nat. Commun.* **14**, 10–17.
- Greber N. D. and Dauphas N. (2019) The chemistry of fine-grained terrigenous sediments reveals a chemically evolved Paleoproterozoic emerged crust. *Geochim. Cosmochim. Acta* **255**, 247–264. Available at: <https://doi.org/10.1016/j.gca.2019.04.012>.
- Greber N. D., Dauphas N., Bekker A., Ptáček M. P., Bindeman I. N. and Hofmann A. (2017) Titanium isotopic evidence for felsic crust and plate tectonics 3.5 billion years ago. *Science (80-.)*. **357**, 1271–1274.
- Hartmann J., Jansen N., Dürr H. H., Kempe S. and Köhler P. (2009) Global CO₂-consumption by chemical weathering: What is the contribution of highly active weathering regions? *Glob. Planet. Change* **69**, 185–194. Available at: <http://dx.doi.org/10.1016/j.gloplacha.2009.07.007>.
- Isson T. T. and Planavsky N. J. (2018) Reverse weathering as a long-term stabilizer of marine pH and planetary climate. *Nature* **560**, 471–475. Available at: <http://dx.doi.org/10.1038/s41586-018-0408-4>.
- Jolicoeur S., Ildefonse P. and Bouchard M. (2000) Kaolinite and Gibbsite Weathering of Biotite within Saprolites and Soils of Central Virginia. *Soil Sci. Soc. Am. J.* **64**, 1118–1129.
- Karathanasis A. D. and Hajek B. F. (1983) Transformation of Smectite to Kaolinite in Naturally Acid Soil Systems: Structural and Thermodynamic Considerations. *Soil Sci. Soc. Am. J.* **47**, 158–163.
- Keller C. B. and Harrison T. M. (2020) Constraining crustal silica on ancient Earth. *Proc. Natl. Acad. Sci. U. S. A.* **117**, 21101–21107.
- Korenaga J., Planavsky N. J. and Evans D. A. D. (2017) Global water cycle and the coevolution of the Earth's interior and surface environment. *Philos. Trans. R. Soc. A Math. Phys. Eng. Sci.* **375**.
- Krissansen-Totton J., Arney G. N. and Catling D. C. (2018) Constraining the climate and ocean pH of the early Earth with a geological carbon cycle model. *Proc. Natl. Acad. Sci.* **115**, 4105–4110. Available at: <https://pnas.org/doi/full/10.1073/pnas.1721296115>.
- Lipp A. G., Shorttle O., Sperling E. A., Brocks J. J., Cole D. B., Crockford P. W., Del Mouro L., Dewing K., Dornbos S. Q., Emmings J. F., Farrell U. C., Jarrett A., Johnson B. W., Kabanov P., Keller C. B., Kunzmann M., Miller A. J., Mills N. T., O'Connell B., Peters S. E., Planavsky N. J., Ritzer S. R., Schoepfer S. D., Wilby P. R. and Yang J. (2021) The composition and weathering of the continents over geologic time. *Geochemical Perspect. Lett.* **17**, 21–26.
- Mackenzie F. T. and Kump L. R. (1995) Reverse Weathering, Clay Mineral Formation, and Oceanic Element Cycles. *Science (80-.)*. **270**, 586–586. Available at: <https://www.science.org/doi/10.1126/science.270.5236.586>.
- Nesbitt H. W. and Young G. M. (1989) Formation and Diagenesis of Weathering Profiles. *J. Geol.* **97**, 129–147. Available at: <https://www.journals.uchicago.edu/doi/10.1086/629290>.
- Opfergelt S. and Delmelle P. (2012) Silicon isotopes and continental weathering processes: Assessing controls on Si transfer to the ocean. *Comptes Rendus - Geosci.* **344**, 723–738. Available at: <http://dx.doi.org/10.1016/j.crte.2012.09.006>.
- Opfergelt S., Georg R. B., Delvaux B., Cabidoche Y. M., Burton K. W. and Halliday A. N. (2012) Silicon isotopes and the tracing of desilication in volcanic soil weathering sequences, Guadeloupe. *Chem. Geol.* **326–327**, 113–122. Available at: <http://dx.doi.org/10.1016/j.chemgeo.2012.07.032>.
- Ptáček M. P., Dauphas N. and Greber N. D. (2020) Chemical evolution of the continental crust from a data-driven inversion of terrigenous sediment compositions. *Earth Planet. Sci. Lett.* **539**.
- Rahman S. and Trower E. J. (2023) Probing surface Earth reactive silica cycling using stable Si isotopes: Mass balance, fluxes, and deep time implications. *Sci. Adv.* **9**, 1–13.
- Roerdink D. L., Ronen Y., Strauss H. and Mason P. R. D. (2022) Emergence of felsic crust and subaerial weathering recorded in Palaeoproterozoic barite. *Nat. Geosci.* **15**, 227–232. Available at: <https://www.nature.com/articles/s41561-022-00902-9>.
- Savage P. S., Georg R. B., Williams H. M. and Halliday A. N. (2013) The silicon isotope composition of the upper continental crust. *Geochim. Cosmochim. Acta* **109**, 384–399. Available at: <http://dx.doi.org/10.1016/j.gca.2013.02.004>.
- Siever R. (1992) The silica cycle in the Precambrian. *Geochim. Cosmochim. Acta* **56**, 3265–3272. Available at: <https://linkinghub.elsevier.com/retrieve/pii/001670379290303Z>.
- Stefurak E. J. T., Lowe D. R., Zentner D. and Fischer W. W. (2015) Sedimentology and geochemistry of Archean silica granules. *Bull. Geol. Soc. Am.* **127**, 1090–1107.
- Tang M., Chen H., Lee C.-T. A. and Cao W. (2024) Subaerial crust emergence hindered by phase-driven lower crust densification on early Earth. *Sci. Adv.* **10**, 1952. Available at: <https://www.science.org>.
- Tosca N. J., Guggenheim S. and Pufahl P. K. (2016) An authigenic origin for Precambrian greenalite: Implications for iron formation and the chemistry of ancient seawater. *Bull. Geol. Soc. Am.* **128**, 511–530.
- Tosca N. J., Macdonald F. A., Strauss J. V., Johnston D. T. and Knoll A. H. (2011) Sedimentary talc in Neoproterozoic carbonate successions. *Earth Planet. Sci. Lett.* **306**, 11–22. Available at:

<http://dx.doi.org/10.1016/j.epsl.2011.03.041>.